# dLLM-Cache: Accelerating Diffusion Large Language Models with Adaptive Caching

## Abstract

Autoregressive Models (ARMs) have long dominated the landscape of Large Language Models. Recently, a new paradigm has emerged in the form of diffusion-based Large Language Models (dLLMs), which generate text by iteratively denoising masked segments. This approach has shown significant advantages and potential. However, dLLMs suffer from high inference latency. Traditional ARM acceleration techniques, such as Key-Value caching, are incompatible with dLLMs due to their bidirectional attention mechanism. To address this specific challenge, our work begins with a key observation that dLLM inference involves a static prompt and a partially dynamic response, where most tokens remain stable across adjacent denoising steps. Based on this, we propose dLLM-Cache, a training-free adaptive caching framework that combines long-interval prompt caching with partial response updates guided by feature similarity. This design enables efficient reuse of intermediate computations without compromising model performance. Extensive experiments on representative dLLMs, including LLaDA 8B and Dream 7B, show that dLLM-Cache achieves up to *9.1×* speedup over standard inference without compromising output quality. Notably, our method brings dLLM inference latency close to that of ARMs under many settings. *Codes are provided in the supplementary material and will be released publicly on GitHub.*

## 1 Introduction

Large language models (LLMs) (Zhao et al., 2023) are foundational to modern AI, powering applications from conversational AI to scientific discovery. While autoregressive models (ARMs) have been the dominant paradigm (Radford, 2018; Brown, 2020; OpenAI, 2022), diffusion-based large language models (dLLMs), such as LLaDA (Nie et al., 2025) and Dream (Ye et al., 2025), have emerged as promising alternatives. These models offer impressive scalability and outperform ARMs in handling challenges like the "reversal curse" (Berglund et al., 2023) due to their bidirectional attention mechanism, demonstrating the potential of diffusion models for complex language tasks.

The practical adoption of dLLMs is hindered by a paradox: despite their potential for parallel decoding, they exhibit a daunting computational complexity of $\mathcal{O}(N^3)$. This inefficiency arises because generating a sequence of length $N$ always requires $N$ denoising iterations in practice, each recalculating bidirectional attention across all tokens without any caching mechanism. This is fundamentally less efficient than standard ARMs, which exploit Key-Value caching (Pope et al., 2023) to reduce the overall computational effort to $\mathcal{O}(N^2)$.

Our work aims to bridge this gap by successfully applying a caching mechanism to dLLMs. To achieve this, we first study two computational redundancies in the inference process of dLLMs as illustrated in Figure 1, which uniform strategies fail to address. First, **prompt redundancy** arises because the input prompt tokens remain constant, yet their internal representations, *e.g.*, attention output, are recomputed in each denoising step. Second, **response dynamics and redundancy** occur as the generated response features evolve. While significant similarity often exists between adjacent steps, suggesting caching potential, not all tokens evolve in the same way. This non-uniform evolution explains why traditional uniform caching strategies are ineffective.

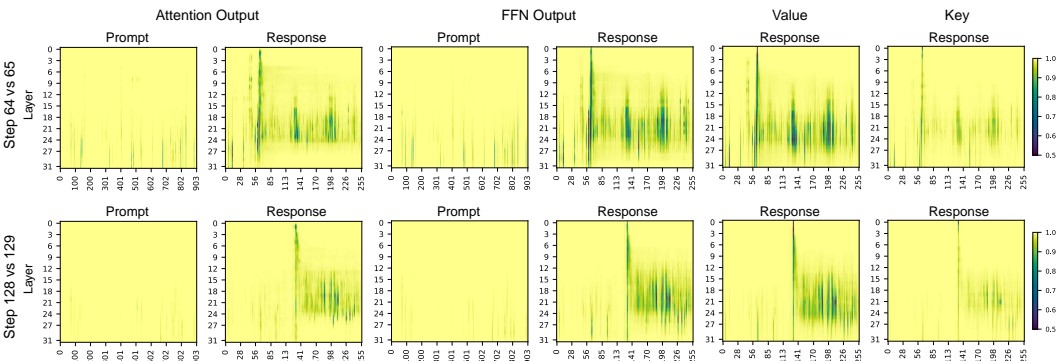

Figure 1: **Cosine similarity of Key, Value, Attention Output and FFN Output between two adjacent denoising steps in a dLLM**, highlighting computational redundancies. The heatmaps show similarity across adjacent steps for prompt and response tokens, respectively, where a lighter color indicates a higher similarity of a token compared with its value in the last step. These results demonstrate: (I) The prompt region exhibits high similarity, while the response region shows different similarity in different tokens. (II) Notably, only a small fraction of response tokens exhibit significantly lower similarity, suggesting that selective recomputation is sufficient. (III) Response tokens' value similarity closely aligns with attention and FFN output similarity, supporting that value changes can serve as an effective indicator to identify those most changed response tokens.

Motivated by these insights, we introduce **dLLM-Cache**, a training-free, adaptive caching mechanism designed to accelerate dLLM inference by exploiting these distinct redundancies. dLLM-Cache employs a differentiated caching strategy comprising two core components:

- **Long-Interval Prompt Caching:** We compute and cache features related to the prompt tokens only at sparse, long intervals, *e.g.*, every 100 steps. These cached features are then reused in all subsequent intermediate steps until the next long interval, drastically reducing the overhead associated with processing the static prompt.
- **Adaptive Short-Interval Response Caching:** Features associated with the response tokens are cached and fully refreshed at more frequent, shorter intervals, *e.g.*, every 10 steps. Between these full refreshes, we adopt an *adaptive partial update* strategy to balance speed and accuracy. Specifically, we identify and selectively update only the most dynamic tokens. As shown in Figure 1, the cosine similarity of a token's Value vector across adjacent steps strongly correlates with changes in its subsequent Attention and FFN Output. This motivates our **V-verify** mechanism, which uses Value similarity as an efficient proxy to select tokens for update.

This differentiated adaptive handling of prompt and response features allows significant inference acceleration while preserving quality, all without retraining. Our main contributions are:

1. We identify and characterize the dual computational redundancies in dLLM inference: quasi-static prompt and dynamic response redundancy.
2. We propose **dLLM-Cache**, a training-free adaptive caching framework that combines long-interval prompt caching with short-interval, similarity-guided partial updates for response tokens.
3. We introduce **V-verify**, a lightweight yet effective mechanism that uses cosine similarity of Value vectors across denoising steps to identify the most changed tokens for partial update, grounded in strong empirical correlation with overall token evolution.
4. We experimentally validate dLLM-Cache across various benchmarks, showing significant inference acceleration, *e.g.*, up to *9.1*× on LLaDA with **lossless impact on response quality**, achieving a superior speed-quality trade-off compared to the baseline and simpler caching methods.

## 2 RELATED WORK

### 2.1 THE LANDSCAPE OF LARGE LANGUAGE MODELS

**Autoregressive Models.** Autoregressive model (ARM) is the dominant paradigm for large language models (LLMs), generating text sequentially via causal attention. These models underpin many state-of-the-art systems (Radford, 2018; Radford et al., 2019; Brown, 2020; OpenAI, 2022).

**Diffusion Models for Language.** Diffusion Models (DMs) (Sohl-Dickstein et al., 2015; Ho et al., 2020; Song et al., 2021) learn to reverse a data corruption process, excelling in continuous domains like images (Rombach et al., 2022; Peebles & Xie, 2023). However, adapting DMs to discrete data like text presents unique challenges, partly due to text's discrete nature. A promising direction involves Masked Diffusion Models (MDMs) (Austin et al., 2021; Lou et al., 2023; Shi et al., 2024; Ou et al., 2024; Zheng et al., 2023; Gong et al., 2024; Nie et al., 2024; He et al., 2022; Reid et al., 2022; Sahoo et al., 2024; Ye et al., 2023), a specific instance of discrete diffusion which operates on discrete sequences by iteratively predicting masked tokens based on their context.

Recent work has scaled MDMs (Nie et al., 2025; Ye et al., 2025), showing performance comparable to ARMs of similar size such as LLaMA3 8B (Dubey et al., 2024). Their bidirectional design helps mitigate limitations specific to ARMs like the reversal curse (Berglund et al., 2023), while extensions to multi-modal (Yang et al., 2025; You et al., 2025) and reasoning tasks (Zhao et al., 2025; Huang et al., 2025; Zhu et al., 2025) further highlight their versatility as a foundation model paradigm.

## 2.2 Acceleration via Caching Mechanisms

**Key–Value Caching in Autoregressive Models.** The most established use of caching in language models is the Key-Value (KV) caching (Pope et al., 2023), which is fundamental to the efficiency of ARMs. In ARMs, causal attention allows for the direct caching of past tokens' key and value states, trading memory for computational speed. However, cache size grows with input length, creating bottlenecks for long-context deployment. To address this, prior work sparsifies caches retrospectively (Xiao et al., 2023; Zhang et al., 2024; Ge et al., 2024; Liu et al., 2023; Li et al., 2025).

**Caching in Diffusion Language Models.** While feature caching has also been explored in ARMs, the bidirectional attention in dLLMs makes traditional KV caching incompatible (Nie et al., 2025), creating a distinct challenge. Concurrent works are beginning to address this gap, but often require cache-aware training (Arriola et al., 2025) or operate under restrictive conditions (Sahoo et al., 2024). Our method, dLLM-Cache, introduces a **training-free** framework that leverages the structure of dLLM inference. It adopts a differentiated caching policy, using infrequent caching for the static prompt and adaptive updates guided by similarity for the dynamic response tokens.

## 3 Methodology

### 3.1 Preliminary

**Training Paradigm of dLLMs.** Unlike the sequential and unidirectional nature of ARMs, dLLMs are trained in a denoising framework that learns to reverse a forward corruption process, where clean sequences are stochastically degraded over a continuous time variable.

Formally, let $\mathbf{x}_0 = (x_1, \ldots, x_L)$ be a clean text sequence sampled from the data distribution $\mathcal{D}$. The forward process defines a continuous time variable $t \in [0, 1]$, with $t = 0$ denoting the clean sequence and $t = 1$ the fully corrupted state. At each time $t$, a corrupted sequence $\mathbf{x}_t$ is produced, where every token $x_{i,0}$ is independently transformed into $x_{i,t}$ according to the rule:

$$x_{i,t} = \begin{cases} [\text{MASK}] & \text{with probability } t \\ x_{i,0} & \text{with probability } 1 - t \end{cases} \tag{1}$$

This per-token independent masking process ensures that as $t \to 1$, the sequence $x_t$ converges to a fully masked sequence.

The model, a bidirectional Transformer parameterized by $\theta$ and denoted $p_\theta$, is trained to reconstruct the original sequence $x_0$ from its corrupted counterpart $x_t$. Training minimizes the negative log-likelihood of the original tokens at masked positions. Let $\mathcal{M}_t$ denote the indices of masked tokens in $x_t$. The loss is defined as:

$$\mathcal{L}(\theta) = -\mathbb{E}_{x_0 \sim \mathcal{D}, t \sim U[0,1]} \left[ \sum_{i \in \mathcal{M}_t} \log p_\theta(x_{i,0}|x_t) \right] \tag{2}$$

This training regimen compels the model to learn a robust representation of language structure by leveraging the full bidirectional context, rather than being constrained by a causal dependency chain.

**Inference Process of dLLMs.** dLLMs generate text via a non-autoregressive process that iteratively denoises a fully masked sequence into the target output. Our work focuses on accelerating this inference procedure. We use LLaDA as a representative example to illustrate it.

Let $\mathcal{T}$ be the token vocabulary and $[\texttt{MASK}] \in \mathcal{T}$ the special mask token. Given a prompt $\mathbf{c} = (c_1, \ldots, c_M)$, the model generates a response $\mathbf{y} = (y_1, \ldots, y_L)$ through $K$ discrete denoising steps, indexed by $k = K$ down to 0. Let $\mathbf{y}^{(k)} \in \mathcal{T}^L$ denote the intermediate state at step $k$, starting from a fully masked sequence:

$$\mathbf{y}^{(K)} = (\underbrace{[\texttt{MASK}], \ldots, [\texttt{MASK}]}_{L \text{ times}}) \tag{3}$$

At each step $k$, a mask predictor $p_\theta$ estimates the distribution over the clean sequence:

$$P_\theta(\mathbf{y}|\mathbf{c}, \mathbf{y}^{(k)}) = p_\theta(\mathbf{c}, \mathbf{y}^{(k)}; \theta) \tag{4}$$

The most likely sequence $\hat{\mathbf{y}}^{(0)}$ is typically obtained via greedy decoding:

$$\hat{\mathbf{y}}^{(0)} = \arg\max_{\mathbf{y} \in \mathcal{T}^L} P_\theta(\mathbf{y}|\mathbf{c}, \mathbf{y}^{(k)}) \tag{5}$$

A transition function $S$ then yields $\mathbf{y}^{(k-1)}$ by selectively updating tokens in $\mathbf{y}^{(k)}$ based on $\hat{\mathbf{y}}^{(0)}$:

$$\mathbf{y}^{(k-1)} = S(\hat{\mathbf{y}}^{(0)}, \mathbf{y}^{(k)}, \mathbf{c}, k) \tag{6}$$

The specific strategy for $S$ may involve confidence-based remasking or semi-autoregressive block updates. While this process enables flexible generation, it incurs high latency due to repeated recomputation across steps, particularly as $K$ grows, as detailed in Appendix A.6.

## 3.2 dLLM-Cache

To alleviate the inference inefficiency of dLLMs, we introduce **dLLM-Cache**, a training-free caching framework. The input prompt remains static across denoising steps, and its internal features are consistently stable, making it suitable for long-interval caching. In contrast, the response sequence evolves dynamically. However, this evolution is highly sparse, as only a small fraction of response tokens change significantly at each step. Such sparsity, evident in Figure 1, suggests that recomputing all response features in every step is often unnecessary.

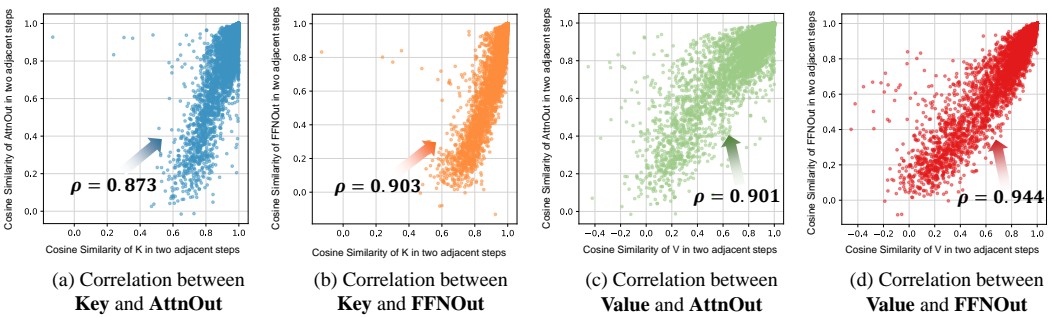

(a) Correlation between **Key** and **AttnOut**

(b) Correlation between **Key** and **FFNOut**

(c) Correlation between **Value** and **AttnOut**

(d) Correlation between **Value** and **FFNOut**

Figure 2: **Correlation of response tokens' K or V changes with other feature changes**. We calculate the cosine similarity between the response tokens' **K** or **V** vectors and their cached counterparts at adjacent steps, select the 25% most dissimilar tokens, and compute the correlation between their similarity with (a) and (c) **AttnOut**, or (b) and (d) **FFNOut** across adjacent steps.

To take advantage of this sparsity, dLLM-Cache selectively updates only a small fraction of response tokens that change most between adjacent steps. The challenge is to identify such tokens efficiently and accurately. Figure 2 reveals that the change in a response token's Value (**V**) or Key (**K**) vector, which is quantified by cosine similarity between current and cached versions, strongly correlates with changes in its subsequent Attention Output (**AttnOut**) and Feedforward Network Output (**FFNOut**). This strong correlation indicates that by monitoring the dynamics of earlier-stage features like **V**, we can effectively identify tokens whose more complex downstream features are also likely to have significantly changed.

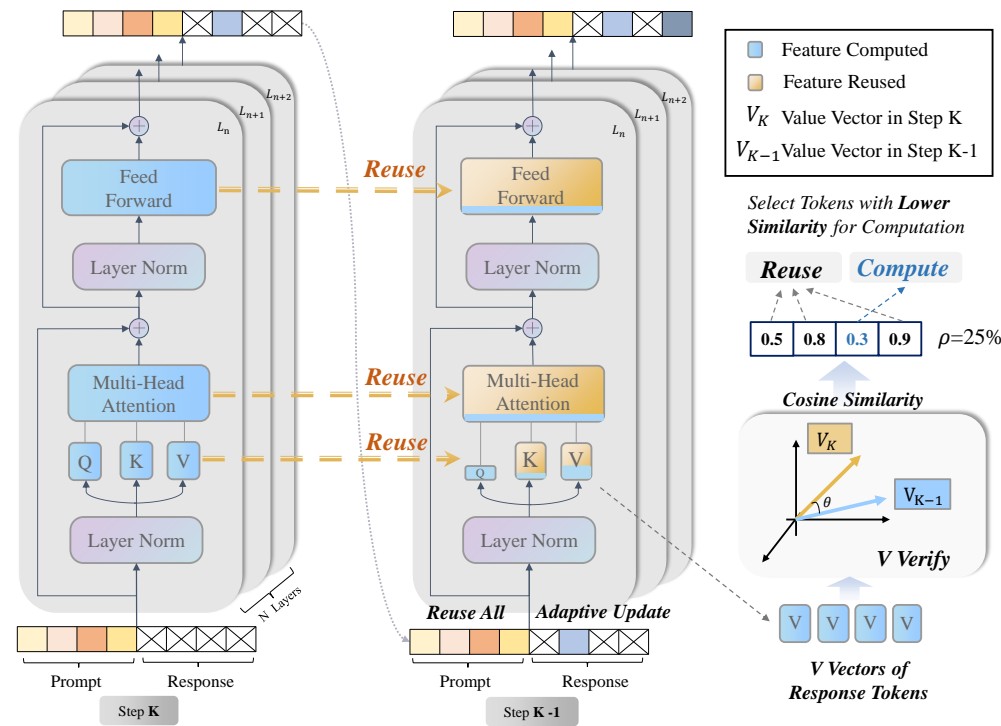

Figure 3: **The dLLM-Cache pipeline.** Prompt features are updated with long intervals, while response features are updated adaptively based on the similarity between newly computed and cached **V** vectors. Response features of tokens with low similarity are updated, and the rest are reused.

Based on this finding, we introduce our **V-verify** mechanism. It uses the cosine similarity between each response token's current **V** vector and its cached counterpart to identify tokens with the largest **V** changes. Only these selected tokens undergo a full feature recomputation and cache update.

Building on this core idea, the overall workflow of dLLM-Cache illustrated in Figure 3 is as follows: For each Transformer layer $l$, we store its $\mathbf{K}^{(l)}$, $\mathbf{V}^{(l)}$, $\mathbf{AttnOut}^{(l)}$, and $\mathbf{FFNOut}^{(l)}$ in a Prompt Cache $\mathcal{C}_p$ and a Response Cache $\mathcal{C}_r$, respectively. Caching is controlled by three hyperparameters: prompt refresh interval $K_p$, response refresh interval $K_r$, and adaptive update ratio $\rho \in [0, 1]$. The inference process generally involves:

**Initialization.** At the very first step ($k = K$), we compute all features from $(\mathbf{c}, \mathbf{y}^{(K)})$. Here, prompt-related features are grouped into $\mathcal{C}_p$, while response-related features go into $\mathcal{C}_r$.

**Iterative Steps.** Next, as $k$ decreases from $K-1$ to $1$, each layer $l$ performs the following operations:
(1) For the prompt, if $k \equiv 0 \pmod{K_p}$, recompute and update $\mathcal{C}_p$; otherwise, reuse.
(2) For the response, if $k \equiv 0 \pmod{K_r}$, fully recompute and update $\mathcal{C}_r$; otherwise, perform adaptive update detailed in Sec. 3.2.2.
(3) Each layer $l$ then continues the forward computation using the available feature version.

**Termination.** The process ends when $k = 0$, producing $\hat{\mathbf{y}}^{(0)}$.

A more compact description of dLLM-Cache is given in Appendix A.10.

### 3.2.1 PROMPT CACHE MANAGEMENT

Since the input prompt $\mathbf{c}$ does not change, its features are largely stable over time. To take advantage of this, dLLM-Cache maintains a Prompt Cache $\mathcal{C}_p$. At $k = K$, all prompt-related features $\mathbf{K}_p^{(l)}, \mathbf{V}_p^{(l)}, \mathbf{AttnOut}_p^{(l)}, \mathbf{FFNOut}_p^{(l)}$ are computed and stored. In subsequent steps, these features are recomputed only every $K_p$ steps; in other steps, they are reused directly from the cache. This reduces the cost of processing the static prompt, particularly when $K_p$ is large.

### 3.2.2 RESPONSE CACHE WITH ADAPTIVE UPDATES

Response features $\mathbf{y}^{(k)}$ evolve over time, though most tokens change gradually, allowing selective updates. The response cache $\mathcal{C}_r$ supports two modes.

**Full Refresh.** All response features are recomputed when $k \equiv 0 \pmod{K_r}$ or $k = K$.

**Adaptive Partial Update.** Otherwise, we first compute the cosine similarity $s_j$ between current and cached Value vectors for each token $j$ (Eq. 7). Then we select the $\lfloor \rho L \rfloor$ tokens with the lowest similarity for updating, recompute their features, and reuse cached values for the rest. Finally, the cache $\mathcal{C}_r$ is updated accordingly.

$$s_j = \frac{(\mathbf{v}_{r,j}^{(l)})^\top \tilde{\mathbf{v}}_{r,j}^{(l)}}{\|\mathbf{v}_{r,j}^{(l)}\| \|\tilde{\mathbf{v}}_{r,j}^{(l)}\|} \tag{7}$$

This adaptive strategy leverages temporal stability to cut computation while preserving accuracy.

## 4 EXPERIMENTS

### 4.1 EXPERIMENT SETTINGS

**Implementation Details.** We evaluated dLLM-Cache on two representative dLLMs: LLaDA 8B (Nie et al., 2025) and Dream 7B (Ye et al., 2025), each with Base and Instruct variants. Following the original inference configurations detailed in Appendix A.9, we conducted our experiments across eight benchmarks. For all models, we applied a fixed adaptive update ratio of $\rho = 0.25$. The prompt refresh interval $K_p$ and response refresh interval $K_r$ are specified in Appendix A.9. All experiments were conducted on the NVIDIA RTX 4090 GPUs.

**Evaluation Metrics.** We evaluate the acceleration and model quality preservation of dLLM-Cache using several metrics. Throughput is measured as Tokens Per Second (TPS), reflecting inference speed. Computational cost is calculated as the average Floating Point Operations (FLOPs) per token. Task performance is assessed using benchmark scores, like accuracy on GSM8K (Cobbe et al., 2021), ensuring dLLM-Cache achieves efficiency gains without compromising model performance. The testing of TPS and FLOPs was performed on a single RTX 4090 GPU.

### 4.2 MAIN RESULTS

**Performance and Efficiency Gains across Models.** Tables 1 and 2 summarize the results for LLaDA 8B and Dream 7B. Across tasks, dLLM-Cache consistently improves inference efficiency without compromising accuracy. On GPQA, for example, applying dLLM-Cache to LLaDA Instruct yields an 8.08× speedup, cutting FLOPs from 22.07T to 2.73T. On GSM8K, Dream Base achieves a 6.90× speedup with no loss in accuracy. Additional exploration of the orthogonality of our dLLM-Cache with recently proposed advanced sampling methods is provided in Appendix A.2.

**Comparison with Contemporary Caching Methods.** We compared dLLM-Cache with two recent cache optimization approaches, dKV-Cache (Ma et al., 2025) and Fast-dLLM (Wu et al., 2025), as shown in Table 3. dLLM-Cache achieves consistently higher throughput across benchmarks. Across benchmarks, dLLM-Cache delivers the highest throughput, reaching 5.33× on GPQA with Dream Base versus 1.74× and 3.83× for the others. Unlike these methods, dLLM-Cache preserves accuracy and generally uses less memory, offering a more practical solution for dLLM inference.

**Comparison with Other Representative LLM.** Table 4 highlights the difference between acceleration strategies. Reducing denoising steps, such as LLaDA 8B Base with 32 steps, raises throughput by 3.63× but drops accuracy to 22.25%. In contrast, applying dLLM-Cache to LLaDA with 128 steps achieves throughput comparable to LLaMA3 8B while retaining 62.32% accuracy, surpassing it by 13.27%. When further combined with SlowFast Sampling (Wei et al., 2025), accuracy improves to 67.17%, showing the orthogonality of our method.

### 4.3 ABLATION STUDY

**Effect of Cache Refresh Interval $K_p$ and $K_r$.** We analyzed how refresh intervals affect efficiency and accuracy. As shown in Figure 4(a), increasing the prompt interval $K_p$ substantially reduces

Table 1: **Comparison of LLaDA 8B with and without dLLM-Cache** on 8 benchmarks.

| Task | Method | Inference Efficiency | | | | Performance |
|---|---|---|---|---|---|---|
| | | TPS↑ | Speed(TPS)↑ | FLOPs(T)↓ | Speed(FLOPs)↑ | Score↑ |
| **Mathematics & Science** | | | | | | |
| GSM8K | LLaDA Base | 7.32 | 1.00× | 16.12 | 1.00× | 69.06 |
| | + dLLM-Cache | $23.19_{+15.87}$ | $3.17\times_{+2.17}$ | $3.21_{-12.91}$ | $5.02\times_{+4.02}$ | $70.66_{+1.60}$ |
| | LLaDA Instruct | 6.95 | 1.00× | 16.97 | 1.00× | 77.48 |
| | + dLLM-Cache | $29.75_{+22.80}$ | $4.28\times_{+3.28}$ | $2.92_{-14.05}$ | $5.81\times_{+4.81}$ | $78.54_{+1.06}$ |
| GPQA | LLaDA Base | 5.12 | 1.00× | 22.97 | 1.00× | 31.91 |
| | + dLLM-Cache | $25.23_{+20.11}$ | $4.93\times_{+3.93}$ | $3.20_{-19.77}$ | $7.18\times_{+6.18}$ | $32.81_{+0.90}$ |
| | LLaDA Instruct | 5.33 | 1.00× | 22.07 | 1.00× | 29.01 |
| | + dLLM-Cache | $28.01_{+22.68}$ | $5.26\times_{+4.26}$ | $2.73_{-19.34}$ | $8.08\times_{+7.08}$ | $29.01_{+0.00}$ |
| Math | LLaDA Base | 8.31 | 1.00× | 14.11 | 1.00× | 30.84 |
| | + dLLM-Cache | $33.92_{+25.61}$ | $4.08\times_{+3.08}$ | $2.61_{-11.50}$ | $5.41\times_{+4.41}$ | $29.84_{-1.00}$ |
| | LLaDA Instruct | 23.65 | 1.00× | 5.16 | 1.00× | 22.32 |
| | + dLLM-Cache | $31.02_{+7.37}$ | $1.31\times_{+0.31}$ | $3.96_{-1.20}$ | $1.30\times_{+0.30}$ | $22.52_{+0.20}$ |
| **General Tasks** | | | | | | |
| MMLU-pro | LLaDA Base | 14.08 | 1.00× | 8.40 | 1.00× | 24.26 |
| | + dLLM-Cache | $45.75_{+31.67}$ | $3.25\times_{+2.25}$ | $2.15_{-6.25}$ | $3.91\times_{+2.91}$ | $24.69_{+0.43}$ |
| | LLaDA Instruct | 14.01 | 1.00× | 8.46 | 1.00× | 36.41 |
| | + dLLM-Cache | $39.63_{+25.62}$ | $2.83\times_{+1.83}$ | $2.62_{-5.84}$ | $3.23\times_{+2.23}$ | $36.08_{-0.33}$ |
| MMLU | LLaDA Base | 8.09 | 1.00× | 14.56 | 1.00× | 63.99 |
| | + dLLM-Cache | $33.52_{+25.43}$ | $4.14\times_{+3.14}$ | $2.64_{-11.92}$ | $5.52\times_{+4.52}$ | $64.26_{+0.27}$ |
| | LLaDA Instruct | 10.12 | 1.00× | 11.85 | 1.00× | 61.24 |
| | + dLLM-Cache | $21.23_{+11.11}$ | $2.10\times_{+1.10}$ | $4.50_{-7.35}$ | $2.63\times_{+1.63}$ | $62.82_{+1.58}$ |
| BBH | LLaDA Base | 6.41 | 1.00× | 18.29 | 1.00× | 44.77 |
| | + dLLM-Cache | $27.90_{+21.49}$ | $4.35\times_{+3.35}$ | $3.09_{-15.20}$ | $5.92\times_{+4.92}$ | $45.04_{+0.27}$ |
| | LLaDA Instruct | 6.18 | 1.00× | 18.98 | 1.00× | 51.49 |
| | + dLLM-Cache | $27.55_{+21.37}$ | $4.46\times_{+3.46}$ | $3.08_{-15.90}$ | $6.16\times_{+5.16}$ | $51.98_{+0.49}$ |
| **Code** | | | | | | |
| MBPP | LLaDA Base | 7.87 | 1.00× | 14.91 | 1.00× | 40.80 |
| | + dLLM-Cache | $24.61_{+16.74}$ | $3.13\times_{+2.13}$ | $3.07_{-11.84}$ | $4.86\times_{+3.86}$ | $40.60_{-0.20}$ |
| | LLaDA Instruct | 7.55 | 1.00× | 15.53 | 1.00× | 39.20 |
| | + dLLM-Cache | $31.73_{+24.18}$ | $4.20\times_{+3.20}$ | $2.80_{-12.73}$ | $5.55\times_{+4.55}$ | $39.60_{+0.40}$ |
| HumanEval | LLaDA Base | 19.98 | 1.00× | 6.03 | 1.00× | 32.92 |
| | + dLLM-Cache | $51.96_{+31.98}$ | $2.60\times_{+1.60}$ | $2.04_{-3.99}$ | $2.96\times_{+1.96}$ | $32.31_{-0.61}$ |
| | LLaDA Instruct | 10.57 | 1.00× | 11.10 | 1.00× | 38.71 |
| | + dLLM-Cache | $44.77_{+34.20}$ | $4.24\times_{+3.24}$ | $2.05_{-9.05}$ | $5.41\times_{+4.41}$ | $39.02_{+0.31}$ |

FLOPs without hurting accuracy, confirming that infrequent prompt updates suffice. Figure 4(b) highlights the importance of response updates. Without prompt caching or adaptive updates ($K_p = 1$, $\rho = 0$, gray line), accuracy drops sharply. In contrast, our setting ($K_p = 50$, $\rho = 0.25$, orange and blue line) maintains high accuracy with much lower cost. This validates our strategy of combining long prompt intervals with short response intervals and adaptive updates. Additional analyses of the Dream model can be found in Appendix A.7.

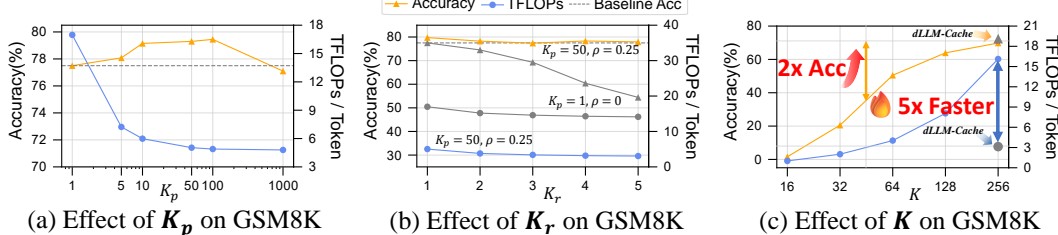

(a) Effect of $K_p$ on GSM8K     (b) Effect of $K_r$ on GSM8K     (c) Effect of $K$ on GSM8K

Figure 4: (a) Varying $K_p$ with $K_r = 1$, $\rho = 0$. (b) Varying $K_r$ under two settings: baseline with $K_p = 1$, $\rho = 0$ in gary and our setup $K_p = 50$, $\rho = 0.25$ in Table 1. (c) Varying denoising steps $K$, where gary patterns are dLLM-Cache with $K = 256$. (a–b) LLaDA Instruct; (c) LLaDA Base.

Table 2: **Comparison of Dream 7B with and without dLLM-Cache** on 8 benchmarks.

| Task | Configuration | Inference Efficiency | | | | Performance |
|---|---|---|---|---|---|---|
| | | TPS↑ | Speed(TPS)↑ | FLOPs(T)↓ | Speed(FLOPs)↑ | Score↑ |
| **Mathematics & Science** | | | | | | |
| GSM8K | Dream Base | 6.36 | 1.00× | 19.59 | 1.00× | 76.95 |
| | + dLLM-Cache | $32.44_{+26.08}$ | $5.10\times_{+4.10}$ | $2.84_{-16.75}$ | $6.90\times_{+5.90}$ | $76.95_{+0.00}$ |
| | Dream Instruct | 6.39 | 1.00× | 19.57 | 1.00× | 77.55 |
| | + dLLM-Cache | $24.52_{+18.13}$ | $3.84\times_{+2.84}$ | $4.24_{-15.33}$ | $4.62\times_{+3.61}$ | $76.80_{-0.75}$ |
| GPQA | Dream Base | 5.80 | 1.00× | 21.66 | 1.00× | 33.92 |
| | + dLLM-Cache | $30.95_{+25.15}$ | $5.33\times_{+4.33}$ | $3.03_{-18.63}$ | $7.15\times_{+6.15}$ | $34.15_{+0.23}$ |
| | Dream Instruct | 5.53 | 1.00× | 22.63 | 1.00× | 34.38 |
| | + dLLM-Cache | $21.98_{+16.45}$ | $3.97\times_{+2.97}$ | $4.69_{-17.94}$ | $4.83\times_{+3.82}$ | $33.93_{-0.45}$ |
| Math | Dream Base | 9.40 | 1.00× | 13.31 | 1.00× | 38.68 |
| | + dLLM-Cache | $36.89_{+27.49}$ | $3.92\times_{+2.92}$ | $2.61_{-10.70}$ | $5.10\times_{+4.10}$ | $37.94_{-0.74}$ |
| | Dream Instruct | 8.85 | 1.00× | 14.11 | 1.00× | 38.28 |
| | + dLLM-Cache | $23.52_{+14.67}$ | $2.66\times_{+1.66}$ | $4.66_{-9.45}$ | $3.03\times_{+2.03}$ | $37.62_{-0.66}$ |
| **General Tasks** | | | | | | |
| MMLU-pro | Dream Base | 15.61 | 1.00× | 7.92 | 1.00× | 24.13 |
| | + dLLM-Cache | $35.86_{+20.25}$ | $2.30\times_{+1.30}$ | $2.89_{-5.03}$ | $2.74\times_{+1.74}$ | $23.86_{-0.27}$ |
| | Dream Instruct | 15.40 | 1.00× | 7.98 | 1.00× | 43.79 |
| | + dLLM-Cache | $23.98_{+8.58}$ | $1.56\times_{+0.56}$ | $4.77_{-3.21}$ | $1.67\times_{+0.67}$ | $43.96_{+0.17}$ |
| MMLU | Dream Base | 9.10 | 1.00× | 13.73 | 1.00× | 73.49 |
| | + dLLM-Cache | $31.07_{+21.97}$ | $3.41\times_{+2.41}$ | $3.27_{-10.46}$ | $4.20\times_{+3.20}$ | $73.20_{-0.29}$ |
| | Dream Instruct | 8.45 | 1.00× | 14.75 | 1.00× | 73.40 |
| | + dLLM-Cache | $38.01_{+29.56}$ | $4.50\times_{+3.50}$ | $2.42_{-12.33}$ | $6.10\times_{+5.10}$ | $73.42_{+0.02}$ |
| BBH | Dream Base | 7.24 | 1.00× | 17.25 | 1.00× | 52.25 |
| | + dLLM-Cache | $29.61_{+22.37}$ | $4.09\times_{+3.09}$ | $3.35_{-13.90}$ | $5.15\times_{+4.15}$ | $51.66_{-0.59}$ |
| | Dream Instruct | 6.98 | 1.00× | 17.90 | 1.00× | 57.07 |
| | + dLLM-Cache | $22.31_{+15.33}$ | $3.20\times_{+2.20}$ | $4.82_{-13.08}$ | $3.71\times_{+2.71}$ | $57.07_{+0.00}$ |
| **Code** | | | | | | |
| MBPP | Dream Base | 8.91 | 1.00× | 14.06 | 1.00× | 54.20 |
| | + dLLM-Cache | $35.69_{+26.78}$ | $4.01\times_{+3.01}$ | $2.66_{-11.40}$ | $5.29\times_{+4.29}$ | $54.20_{+0.00}$ |
| | Dream Instruct | 8.46 | 1.00× | 14.65 | 1.00× | 57.00 |
| | + dLLM-Cache | $29.77_{+21.31}$ | $3.52\times_{+2.52}$ | $3.33_{-11.32}$ | $4.40\times_{+3.40}$ | $56.80_{-0.20}$ |
| HumanEval | Dream Base | 21.43 | 1.00× | 5.68 | 1.00× | 58.53 |
| | + dLLM-Cache | $27.40_{+5.97}$ | $1.28\times_{+0.28}$ | $4.17_{-1.51}$ | $1.36\times_{+0.36}$ | $57.31_{-1.22}$ |
| | Dream Instruct | 17.88 | 1.00× | 6.84 | 1.00× | 57.92 |
| | + dLLM-Cache | $28.03_{+10.15}$ | $1.57\times_{+0.57}$ | $3.94_{-2.90}$ | $1.74\times_{+0.74}$ | $56.09_{-1.83}$ |

Table 3: Comparison of LLaDA (left) and Dream (right) with different caching methods.

| Task | Method | TPS↑ | Speed↑ | Memory↓ | Score↑ | Task | Method | TPS↑ | Speed↑ | Memory↓ | Score↑ |
|---|---|---|---|---|---|---|---|---|---|---|---|
| GSM8K | LLaDA Instruct | 6.95 | 1.00× | 15.86 | 77.48 | GSM8K | Dream Base | 6.36 | 1.00× | 15.73 | 76.95 |
| | + dKV-Cache | 8.89 | 1.28× | 21.08 | **79.30** | | + dKV-Cache | 10.26 | 1.61× | **16.14** | 76.57 |
| | + Fast-dLLM | 19.11 | 2.75× | 19.48 | 75.89 | | + Fast-dLLM | 21.36 | 2.08× | 19.95 | 74.30 |
| | + dLLM-Cache | **29.75** | **4.28×** | 17.85 | 78.54 | | + dLLM-Cache | **32.44** | **5.10×** | 16.76 | **76.95** |
| MMLU | LLaDA Instruct | 10.12 | 1.00× | 15.54 | 61.24 | GPQA | Dream Base | 5.80 | 1.00× | 15.77 | 33.92 |
| | + dKV-Cache | 14.34 | 1.42× | 17.88 | 60.87 | | + dKV-Cache | 10.11 | 1.74× | **16.23** | 32.83 |
| | + Fast-dLLM | 20.51 | 2.03× | 17.13 | 61.43 | | + Fast-dLLM | 22.23 | 3.83× | 20.69 | 31.31 |
| | + dLLM-Cache | **21.23** | **2.10×** | **16.61** | **62.82** | | + dLLM-Cache | **30.95** | **5.33×** | 16.93 | **34.15** |
| HumanEval | LLaDA Instruct | 10.57 | 1.00× | 15.39 | 38.71 | MMLU | Dream Base | 9.10 | 1.00× | 15.64 | 73.49 |
| | + dKV-Cache | 14.40 | 1.36× | 17.17 | 37.20 | | + dKV-Cache | 12.80 | 1.41× | **15.92** | 72.77 |
| | + Fast-dLLM | 21.50 | 2.03× | **16.60** | 36.59 | | + Fast-dLLM | 23.69 | 2.60× | 18.32 | 72.69 |
| | + dLLM-Cache | **44.77** | **4.24×** | 16.65 | **39.02** | | + dLLM-Cache | **31.07** | **3.41×** | 16.37 | **73.20** |

**Effect of Update Ratio $\rho$ and Selection Strategy.** We investigated how different token selection strategies impact performance under varying adaptive update ratios $\rho$. Figure 5 reports accuracy and computational cost on GSM8K when using three strategies: **V-verify**, **K-verify**, and random selection. Both similarity-based strategies consistently outperform random selection across a wide range of $\rho$ values, confirming the importance of dynamic, feature-driven updates. In particular, value-based selection achieves the highest accuracy around $\rho = 0.25$, while requiring significantly fewer FLOPs than full recomputation. This suggests that moderate, targeted updates, *e.g.*, $\rho \approx 0.25$ strike a favorable trade-off between efficiency and output quality.

Table 4: **Comparison of LLaDA 8B Base with other representative LLM** on GSM8K.

| Method | Steps | Throughput(TPS)↑ | Speed ↑ | Accuracy(%)↑ | Memory (GB)↓ |
|---|---|---|---|---|---|
| LLaMA3 8B | - | 47.73 | 1.00× | 49.05 | 16.06 |
| LLaDA Base | 128 | 14.77$_{-32.96}$ | 1.00× | 64.14$_{+15.09}$ | 16.94 |
| LLaDA Base | 32 | 53.55$_{+5.82}$ | 3.63×$_{+2.63}$ | 22.25$_{-26.80}$ | 16.94 |
| + Cache | 128 | 49.15$_{+1.42}$ | 3.33×$_{+2.33}$ | 62.32$_{+13.27}$ | 17.93 |
| + Cache + SlowFast | - | 49.86$_{+2.13}$ | 3.38×$_{+2.33}$ | 67.17$_{+18.12}$ | 17.93 |

## 5 DISCUSSION

**Effect of Denoising Steps.** In dLLMs, the number of denoising steps determines a trade-off between quality and latency. Increasing the steps improves output accuracy but also raises inference cost, as shown in Figure 4(c). Simply reducing the steps accelerates inference but causes severe performance degradation. On GSM8K, dLLM-Cache achieves a 5× lossless speedup at 256 steps, matching the computational cost of a baseline with only 48 steps while more than doubling its accuracy. This shows that our method achieves both efficiency and quality, unlike simple step reduction.

**Storage Overhead of Caching.** dLLM-Cache stores four types of intermediate features per layer: **K**, **V**, **AttnOut**, and **FFNOut**. The total cache size scales with the number of tokens $T$, embedding dimension $d$, and number of layers $L$, giving a cost of $T \times d \times 4 \times L$ as detailed in Appendix A.8. Since only one version per layer is cached, the overall footprint remains stable. As shown in Table 4, on GSM8K with LLaDA 8B Base, peak GPU usage is 16.94GB without caching, 17.93GB with dLLM-Cache, and 16.06GB for LLaMA3 8B. This small 5% memory increase yields up to 9× acceleration, making it a favorable tradeoff.

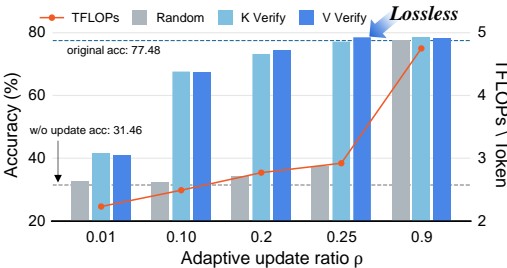

Figure 5: **Effect of token selection strategy** on GSM8K using LLaDA 8B Instruct model under varying update ratios $\rho$.

Figure 6: **TPS versus $\rho$.** A notable decrease in TPS at minimal $\rho$ reflects the fixed cost of initiating selective updates.

**Cost of V-verify and the Fixed Update Overheads.** Our **V-verify** mechanism uses lightweight **V** vector similarity for identifying dynamic tokens. While **V-verify** itself is computationally inexpensive, as illustrated in Figure 6, practical speedup from adaptive partial updates is constrained by fixed operational overheads. Figure 6 shows a notable decrease in TPS as the update ratio $\rho$ approaches zero. This base cost arises because initiating any selective recomputation ($\rho > 0$) triggers non-negligible system-level latencies, *e.g.*, for GPU kernel management and data movement that are not strictly proportional to the number of updated tokens. Consequently, at very low $\rho$ values, these fixed overheads dominate, limiting further run time savings. An optimal $\rho$ must balance these fixed costs against saved dynamic computation, while preserving model quality. Figure 5 suggests $\rho \approx 0.25$ offers an effective trade-off between the costs of activating selective updates and the benefits of reduced computation, optimizing overall efficiency and fidelity.

## 6 CONCLUSION

We present dLLM-Cache, a training-free and model-agnostic caching method for accelerating inference in diffusion-based large language models. Extensive experiments on LLaDA and Dream show that dLLM-Cache achieves up to *9.1×* speedup without compromising generation quality.

## REPRODUCIBILITY STATEMENT

To ensure reproducibility, we have included the source code in the supplementary materials.

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

# A  APPENDIX

## A.1  THE USE OF LARGE LANGUAGE MODELS (LLMS)

During the preparation of this manuscript, we utilized a large language model to aid and polish the writing. The LLM served as a general-purpose assistant for improving grammar, clarity, and phrasing. All content was reviewed and edited by the authors.

## A.2  COMPATIBILITY WITH ADVANCED SAMPLING METHODS.

Our dLLM-Cache is orthogonal to recent sampling-based acceleration methods, such as SlowFast Sampling (Wei et al., 2025). When combined, as shown in Table 5, the two methods achieve greater inference speedups while preserving model performance.

Table 5: **Performance of LLaDA Base with dLLM-Cache and SlowFast Sampling.**

| Task | Method | Inference Efficiency | | Performance |
|---|---|---|---|---|
| | | TPS↑ | Speed(TPS)↑ | Score↑ |
| **Mathematics & Science** | | | | |
| GSM8K | LLaDA Base | 4.55 | 1.00× | 69.83 |
| | Sampling + Cache | 26.99$_{+22.44}$ | 5.93×$_{+4.93}$ | 69.60$_{-0.23}$ |
| GPQA | LLaDA Base | 3.31 | 1.00× | 31.47 |
| | Sampling + Cache | 29.06$_{+25.75}$ | 8.78×$_{+7.78}$ | 33.48$_{+2.01}$ |
| Math | LLaDA Base | 5.14 | 1.00× | 30.16 |
| | Sampling + Cache | 26.50$_{+21.36}$ | 5.16×$_{+4.16}$ | 29.42$_{-0.74}$ |
| **General Tasks** | | | | |
| MMLU-pro | LLaDA Base | 9.16 | 1.00× | 23.30 |
| | Sampling + Cache | 33.38$_{+24.22}$ | 3.64×$_{+2.64}$ | 25.53$_{+2.23}$ |
| MMLU | LLaDA Base | 5.02 | 1.00× | 62.11 |
| | Sampling + Cache | 38.42$_{+33.40}$ | 7.65×$_{+6.65}$ | 61.20$_{-0.91}$ |
| BBH | LLaDA Base | 4.04 | 1.00× | 44.97 |
| | Sampling + Cache | 36.04$_{+32.00}$ | 8.92×$_{+7.92}$ | 44.81$_{-0.16}$ |
| **Code** | | | | |
| MBPP | LLaDA Base | 4.98 | 1.00× | 40.80 |
| | Sampling + Cache | 27.26$_{+22.28}$ | 5.47×$_{+3.87}$ | 39.00$_{-1.80}$ |
| HumanEval | LLaDA Base | 11.24 | 1.00× | 31.71 |
| | Sampling + Cache | 41.14$_{+29.90}$ | 3.66×$_{+2.66}$ | 31.10$_{-0.61}$ |

## A.3  EFFECTIVENESS ON LONG-PROMPT SCENARIOS.

The benefits of dLLM-Cache are particularly pronounced in scenarios involving long input prompts, common in tasks like document-based question answering. Our Long-Interval Prompt Caching mechanism significantly curtails redundant computations for the extensive static prompt portion by refreshing its cache only at long intervals. For instance, when applying dLLM-Cache to the LLaDA 8B Base model on the LongBench-HotpotQA (Bai et al., 2023) task, we not only achieved a **9.1×** **speedup** over the unaccelerated baseline but also observed a performance improvement, with the F1 score increasing from 34.56 to 36.10. This highlights the particular suitability of dLLM-Cache for dLLM applications requiring extensive contextual understanding, where our caching strategy for long static prompts can be maximally leveraged.

## A.4  PERFORMANCE ANALYSIS ON LONG AND SEMANTICALLY DIVERSE PROMPTS

To comprehensively evaluate the applicability of dLLM-Cache in more challenging, real-world scenarios, we conducted a thorough set of experiments on the LongBench benchmark. LongBench is designed to test model capabilities on long-context tasks and includes six major categories:

Table 6: **Comparison of LongBench performance on LLaDA Instruct and Dream Instruct with and without dLLM-Cache.**

| Method | Single-Doc. QA | | Multi-Doc. QA | | | Summarization | | | Few-shot Learning | | | Synthetic | Code | | Ave. Score |
|---|---|---|---|---|---|---|---|---|---|---|---|---|---|---|---|
| | Qasper | MF-en | HotpotQA | 2WikiMQA | Musique | GovReport | QMSum | MultiNews | TREC | TriviaQA | SAMSum | PRe | Lcc | RB-P | |
| LLaDA Instruct | 16.96 | 31.31 | 14.68 | 17.60 | 11.48 | 29.24 | 21.93 | 27.58 | 65.20 | 47.98 | 40.51 | 98.17 | 65.69 | 59.57 | **39.14** |
| + dLLM-Cache | 15.26 | 29.62 | 13.87 | 17.17 | 10.44 | 29.75 | 22.06 | 26.68 | 66.00 | 44.94 | 41.86 | 97.44 | 66.07 | 59.34 | **38.61** |
| Dream Instruct | 28.17 | 36.23 | 27.65 | 32.43 | 11.83 | 5.04 | 14.29 | 5.95 | 73.00 | 89.25 | 37.84 | 16.92 | 38.91 | 45.08 | **33.04** |
| + dLLM-Cache | 26.55 | 39.86 | 27.66 | 32.09 | 11.12 | 4.40 | 13.89 | 5.51 | 73.50 | 89.59 | 36.07 | 12.05 | 39.88 | 45.57 | **32.70** |

single-document QA, multi-document QA, summarization, few-shot learning, synthetic tasks, and code completion. The benchmark is notable for its exceptionally long texts and its high degree of semantic and structural diversity, making it an effective measure of model performance on complex, long-context inputs.

We evaluated both the LLaDA Instruct and Dream Instruct models, comparing their performance with and without dLLM-Cache enabled. The detailed results are presented in Table 6. As the results demonstrate, the average score for LLaDA Instruct with dLLM-Cache is 38.61, which is highly comparable to the baseline score of 39.14. Similarly, for Dream Instruct, the average score with the cache enabled is 32.70, showing strong performance retention against the baseline of 33.04. These results, spanning a wide range of tasks that require deep semantic understanding and long-range dependency reasoning, confirm the robust performance of our caching strategy.

### A.5 IMPACT OF SIMILARITY METRIC.

We compared cosine similarity and L2 distance as similarity metrics for **V-verify**. On GSM8K with LLaDA 8B Instruct, cosine similarity achieved 78.54% accuracy, significantly outperforming L2 distance at 55.95%. This shows that cosine similarity better captures semantic change, and we adopt it as the default throughout our method.

### A.6 COMPLEXITY AND LATENCY ANALYSIS

In this section, we provide a detailed computational complexity analysis for the original dLLM inference process and our proposed dLLM-Cache framework.

**Complexity of the Original dLLM Model.** Standard dLLMs, such as LLaDA and Dream, utilize a multi-layer Transformer architecture with **bidirectional attention**. Text generation is performed over **K** iterative denoising steps, starting from a fully masked sequence. At each step, the model executes a full forward pass over the entire input sequence of length $n$. The per-step computational cost, measured in FLOPs, is dominated by the attention and feed-forward network (FFN) layers:

$$\text{FLOPs}_{\text{step}} = T \cdot (8nd^2 + 4n^2d + 4ndm) \tag{8}$$

where $T$ is the number of Transformer layers, $n$ is the sequence length, $d$ is the hidden dimension size, and $m$ is the intermediate size of the FFN.

Consequently, the total inference complexity for a standard dLLM is the per-step cost multiplied by the number of steps $K$:

$$\text{FLOPs}_{\text{dLLM}} = K \cdot T \cdot (8nd^2 + 4n^2d + 4ndm) \tag{9}$$

**Complexity with dLLM-Cache.** dLLM-Cache optimizes this process by caching intermediate states and selectively updating only a fraction of tokens. This partitions the computation into three main types: full refreshes, response-only refreshes, and adaptive partial updates. The total complexity

can be approximated as:

$$
\begin{aligned}
\text{FLOPs}_{\text{dLLM-Cache}} \approx \; & \frac{K}{K_p} \cdot T \cdot (8nd^2 + 4n^2 d + 4ndm) \\
& + \left( \frac{K}{K_r} - \frac{K}{K_p} \right) \cdot T \cdot (8rd^2 + 4rnd + 4rdm) \\
& + K \cdot \left( 1 - \frac{1}{K_r} \right) \cdot T \cdot (8\hat{r}d^2 + 4\hat{r}nd + 4\hat{r}dm)
\end{aligned}
\tag{10}
$$

where $K_p$ and $K_r$ are the refresh intervals for the prompt and response, respectively; $p$ and $r$ are the prompt and response lengths ($n = p + r$); and $\hat{r} = \rho \cdot r$ is the number of updated response tokens during adaptive steps, with $\rho$ being the adaptive update ratio.

The first term represents the cost of full refreshes occurring every $K_p$ steps. The second term accounts for the periodic response-only refreshes. The final, and most frequent, term reflects the cost of lightweight adaptive updates applied only to the $\hat{r}$ most dynamic response tokens.

**Computation Savings.** The primary source of acceleration in dLLM-Cache comes from replacing the expensive quadratic attention term, $4n^2 d$, with a much smaller term, $4\hat{r}nd$, for the majority of the denoising steps. The relative computational savings can be expressed as:

$$
\text{Savings} = 1 - \frac{\text{FLOPs}_{\text{dLLM-Cache}}}{\text{FLOPs}_{\text{dLLM}}}
\tag{11}
$$

As demonstrated in our experiments, this significant reduction in computational demand leads to substantial improvements in inference speed, achieving up to a $9.1\times$ speedup in practical scenarios.

### A.7 DETAILED SENSITIVITY ANALYSIS ON DREAM 7B

As demonstrated in the main paper, dLLM-Cache is effective across different dLLM architectures, including both LLaDA and Dream. This highlights the generalizability of our approach, which targets computational redundancies fundamental to the diffusion process rather than model-specific artifacts.

To further substantiate the robustness of our method and provide deeper insight into its behavior, this section presents a detailed sensitivity analysis of dLLM-Cache's key hyperparameters when applied to the Dream 7B model. The results, shown in Table 7, Table 8, and Table 9, reveal performance trends that are highly consistent with those observed for LLaDA. This confirms the stable and predictable behavior of our method across different models.

Table 7: Sensitivity analysis of the adaptive update ratio $\rho$ on Dream 7B for the GPQA benchmark. Hyperparameters are set to $K_p = 25$ and $K_r = 4$.

| $\rho$ | 0 | 0.1 | 0.2 | 0.25 | 0.3 | 0.5 | 0.75 | 1 |
|---|---|---|---|---|---|---|---|---|
| Accuracy (%) | 35.04 | 36.16 | 35.93 | 35.04 | 35.04 | 34.59 | 35.49 | 35.26 |

Table 8: Sensitivity analysis of the prompt refresh interval $K_p$ on Dream 7B for the GPQA benchmark. Hyperparameters are set to $K_r = 4$ and $\rho = 0.25$.

| $K_p$ | 10 | 25 | 50 | 100 |
|---|---|---|---|---|
| Accuracy (%) | 35.04 | 35.04 | 35.04 | 35.04 |

Table 9: Sensitivity analysis of the response refresh interval $K_r$ on Dream 7B for the GPQA benchmark. Hyperparameters are set to $K_p = 25$ and $\rho = 0.25$.

| $K_r$ | 2 | 4 | 6 |
|---|---|---|---|
| Accuracy (%) | 36.16 | 35.04 | 33.92 |

## A.8 Proof of Storage Overhead of Caching

*Theorem: The storage overhead of caching in our method is $O(T \times d \times 4 \times L)$, where $T$ is the number of tokens, $d$ is the embedding dimension, and $L$ is the number of layers.*

*Proof.* We first define the memory required for each layer of the model. In our method, four types of intermediate features are stored per layer: **K**, **V**, **AttnOut**, and **FFNOut**. Each feature has a size of $T \times d$, where $T$ is the number of tokens and $d$ is the embedding dimension.

Let $M_{\text{layer}}$ denote the memory required for each layer. Since four feature types are cached per layer, the memory required for one layer is:

$$M_{\text{layer}} = 4 \times T \times d$$

This accounts for the four different feature types stored per token in the layer.

Now, consider the entire model, which consists of $L$ layers. The total memory required for caching all layers is simply the memory required for one layer multiplied by the number of layers:

$$M_{\text{total}} = L \times M_{\text{layer}} = L \times 4 \times T \times d$$

Next, we consider the precision used to store these features. In our method, we use bfloat16 precision, where each element requires 2 bytes of memory. Therefore, the total memory required for storing all features in terms of bytes is:

$$M_{\text{total}} = 2 \times L \times 4 \times T \times d \text{ bytes}$$

Finally, in asymptotic analysis, we focus on the growth rate of the memory overhead and ignore constant factors such as the factor of 2 bytes for precision. Therefore, the storage overhead grows as:

$$O(T \times d \times 4 \times L)$$

This completes the proof. $\square$

## A.9 Experimental Details

This section provides the detailed configuration settings used in our experiments across a variety of tasks for both the Instruct and Base variants of the evaluated diffusion-based large language models. For each task, we report the number of denoising steps, the block length, the total generation length, the remasking strategy, the number of few-shot examples used (if any), the prompt refresh interval $K_p$, and the response refresh interval $K_r$. All models use the low-confidence remasking strategy unless otherwise specified.

The values of $K_p$ and $K_r$ can be flexibly adjusted according to task requirements rather than through hyperparameter tuning. For example, in applications that are sensitive to accuracy, such as code generation or mathematical reasoning, smaller values of $K_p$ and $K_r$ may be preferred to ensure higher fidelity. In contrast, in applications that emphasize efficiency, such as casual dialogue, larger values can be adopted to reduce computational overhead. It is worth noting that our method does not rely on tuning $K_p$ and $K_r$ for performance gains; instead, these intervals simply reflect task-specific trade-offs between efficiency and precision.

The magnitude of gains sometimes varies across Base and Instruct models due to benchmark configurations from prior work (Nie et al., 2025). For example, MMLU uses a 256-token generation length and decoding steps for Base but only 3 for Instruct, leading to different speedup ratios since our acceleration scales with the number of tokens and denoising steps, as detailed in Appendix A.6.

Table 10: Experimental settings for Instruct model across selected benchmarks.

| Task | Steps | Block Len | Gen Len | Few-shot |
|---|---|---|---|---|
| GSM8K | 256 | 8 | 256 | 4 |
| GPQA | 128 | 64 | 128 | 5 |
| Math | 256 | 256 | 256 | 0 |
| MMLU-pro | 256 | 256 | 256 | 0 |
| MMLU | 3 | 3 | 3 | 5 |
| MBPP | 512 | 32 | 512 | 3 |
| BBH | 256 | 256 | 256 | 3 |
| HumanEval | 512 | 32 | 512 | 0 |

Table 11: Interval steps for LLaDA Base across selected benchmarks.

| | GSM8K | GPQA | Math | MMUL-pro | MMLU | BBH | MBPP | HumanEval | Avg. |
|---|---|---|---|---|---|---|---|---|---|
| $K_p$ | 25 | 100 | 50 | 100 | 100 | 50 | 25 | 100 | 69 |
| $K_r$ | 5 | 8 | 8 | 6 | 6 | 6 | 4 | 5 | 6 |

Table 12: Interval steps for LLaDA Instruct across selected benchmarks.

| | GSM8K | GPQA | Math | MMUL-pro | MMLU | BBH | MBPP | HumanEval | Avg. |
|---|---|---|---|---|---|---|---|---|---|
| $K_p$ | 50 | 50 | 50 | 51 | 100 | 100 | 100 | 25 | 66 |
| $K_r$ | 7 | 6 | 1 | 3 | 7 | 5 | 5 | 5 | 5 |

Table 13: Interval steps for Dream Base across selected benchmarks.

| | GSM8K | GPQA | Math | MMUL-pro | MMLU | BBH | MBPP | HumanEval | Avg. |
|---|---|---|---|---|---|---|---|---|---|
| $K_p$ | 100 | 100 | 100 | 25 | 100 | 25 | 25 | 5 | 60 |
| $K_r$ | 8 | 8 | 4 | 2 | 2 | 4 | 8 | 1 | 5 |

Table 14: Interval steps for Dream Instruct across selected benchmarks.

| | GSM8K | GPQA | Math | MMUL-pro | MMLU | BBH | MBPP | HumanEval | Avg. |
|---|---|---|---|---|---|---|---|---|---|
| $K_p$ | 25 | 10 | 50 | 5 | 100 | 10 | 10 | 50 | 33 |
| $K_r$ | 2 | 8 | 1 | 1 | 8 | 2 | 8 | 1 | 4 |

## A.10 CORE ALGORITHMIC WORKFLOW OF DLLM-CACHE

Algorithm 1 outlines the full forward computation process of dLLM-Cache, our training-free adaptive caching framework for diffusion-based large language models. At each denoising step, the algorithm dynamically determines whether to refresh prompt and/or response features based on predefined cache intervals ($K_p$ for prompt, $K_r$ for response). When neither full refresh condition is met, dLLM-Cache employs an adaptive update mechanism that selectively recomputes features for response tokens exhibiting the most significant semantic drift, as measured by value vector similarity. This selective caching strategy enables substantial computational savings without compromising generation quality, and is compatible with arbitrary Transformer-based denoising networks.

---

**Algorithm 1** dLLM-Cache: Main Inference Algorithm

---

**Require:** Prompt $\mathbf{c}$, initial masked sequence $\mathbf{y}^{(K)}$, denoising steps $K$, prompt refresh interval $K_p$, response refresh interval $K_r$, adaptive update ratio $\rho$

**Ensure:** Final prediction $\hat{\mathbf{y}}^{(0)}$

1: /* Initialize caches at step $k = K$ */
2: $\mathcal{C}_p, \mathcal{C}_r \leftarrow$ InitializeCache($\mathbf{c}, \mathbf{y}^{(K)}$)                       ▷ Algorithm 2
3: Generate prediction $\hat{\mathbf{y}}^{(0)}$ using model $f_\phi$          ▷ Needs initial pass or separate handling
4: $\mathbf{y}^{(K-1)} \leftarrow S(\hat{\mathbf{y}}^{(0)}, \mathbf{y}^{(K)}, \mathbf{c}, K)$
5: **for** $k = K - 1$ **down to** 1 **do**
6:     $\mathbf{x}_{layer\_in} \leftarrow [\mathbf{c}; \mathbf{y}^{(k)}]$                       ▷ Initial input for layer 1 at step k
7:     **for** each layer $l$ in the Transformer network **do**
8:         /* Determine refresh conditions based on intervals */
9:         refresh_prompt $\leftarrow (k \bmod K_p = 0)$                       ▷ Refresh prompt every $K_p$ steps
10:         refresh_response $\leftarrow (k \bmod K_r = 0)$                       ▷ Refresh response every $K_r$ steps
11:         /* Cache usage strategy based on refresh conditions */
12:         **if** refresh_prompt **and** refresh_response **then**
13:             $\mathbf{x}_{layer\_out}, \mathcal{C}_p, \mathcal{C}_r \leftarrow$ FullRefresh($\mathbf{x}_{layer\_in}, l, \mathcal{C}_p, \mathcal{C}_r$)                       ▷ Algorithm 3
14:         **else if** refresh_prompt **and not** refresh_response **then**
15:             $\mathbf{x}_{layer\_out}, \mathcal{C}_p, \mathcal{C}_r \leftarrow$ RefreshPromptOnly($\mathbf{x}_{layer\_in}, l, \mathcal{C}_p, \mathcal{C}_r$)                       ▷ Algorithm 4
16:         **else if not** refresh_prompt **and** refresh_response **then**
17:             $\mathbf{x}_{layer\_out}, \mathcal{C}_p, \mathcal{C}_r \leftarrow$ RefreshResponseOnly($\mathbf{x}_{layer\_in}, l, \mathcal{C}_p, \mathcal{C}_r$)                       ▷ Algorithm 5
18:         **else**
19:             /* When neither needs full refresh */
20:             $\mathbf{x}_{layer\_out}, \mathcal{C}_p, \mathcal{C}_r \leftarrow$ AdaptiveUpdate($\mathbf{x}_{layer\_in}, l, \mathcal{C}_p, \mathcal{C}_r, \rho$)                       ▷ Algorithm 6
21:         **end if**
22:         $\mathbf{x}_{layer\_in} \leftarrow \mathbf{x}_{layer\_out}$                       ▷ Update input for the next layer
23:     **end for**                       ▷ End layer loop
24:     Generate prediction $\hat{\mathbf{y}}^{(0)}$ using model $f_\phi$ with final layer output $\mathbf{x}_{layer\_out}$
25:     $\mathbf{y}^{(k-1)} \leftarrow S(\hat{\mathbf{y}}^{(0)}, \mathbf{y}^{(k)}, \mathbf{c}, k)$                       ▷ Apply transition function
26: **end for**                       ▷ End step loop
27: **return** final prediction $\hat{\mathbf{y}}^{(0)}$

---

---

**Algorithm 2** dLLM-Cache: Cache Structure and Initialization

---

**Require:** Prompt $\mathbf{c}$, initial masked sequence $\mathbf{y}^{(K)}$, Transformer network with $L$ layers

1: /* Cache Structure Definition */
2: **for** layer $l \in \{1, 2, \ldots, L\}$ **do**
3:   $\mathcal{C}_p[l][\texttt{kv\_cache}] \leftarrow \{\}$        ▷ Prompt key-value cache
4:   $\mathcal{C}_p[l][\texttt{attn}] \leftarrow \{\}$        ▷ Prompt attention output cache
5:   $\mathcal{C}_p[l][\texttt{mlp}] \leftarrow \{\}$        ▷ Prompt FFN output cache
6:   $\mathcal{C}_r[l][\texttt{kv\_cache}] \leftarrow \{\}$        ▷ Response key-value cache
7:   $\mathcal{C}_r[l][\texttt{attn}] \leftarrow \{\}$        ▷ Response attention output cache
8:   $\mathcal{C}_r[l][\texttt{mlp}] \leftarrow \{\}$        ▷ Response FFN output cache
9: **end for**
10: /* Initial Caching (Step $k = K$) */
11: $\mathbf{x}_{in} \leftarrow [\mathbf{c}; \mathbf{y}^{(K)}]$        ▷ Concatenated input for the first layer
12: **for** layer $l \in \{1, 2, \ldots, L\}$ **do**
13:   /* --- Attention Block --- */
14:   $\mathbf{x}_{norm} \leftarrow \text{LayerNorm}(\mathbf{x}_{in})$
15:   $\mathbf{Q}, \mathbf{K}, \mathbf{V} \leftarrow \text{Q\_proj}(\mathbf{x}_{norm}), \text{K\_proj}(\mathbf{x}_{norm}), \text{V\_proj}(\mathbf{x}_{norm})$
16:   /* Split K, V for caching */
17:   $\mathbf{K}_p, \mathbf{K}_r \leftarrow \mathbf{K}_{1:|\mathbf{c}|}, \mathbf{K}_{|\mathbf{c}|+1:}$
18:   $\mathbf{V}_p, \mathbf{V}_r \leftarrow \mathbf{V}_{1:|\mathbf{c}|}, \mathbf{V}_{|\mathbf{c}|+1:}$
19:   $\mathcal{C}_p[l][\texttt{kv\_cache}] \leftarrow \{\mathbf{K}_p, \mathbf{V}_p\}$        ▷ Store prompt KV
20:   $\mathcal{C}_r[l][\texttt{kv\_cache}] \leftarrow \{\mathbf{K}_r, \mathbf{V}_r\}$        ▷ Store response KV
21:   $\mathbf{AttnOut} \leftarrow \text{Attention}(\mathbf{Q}, \mathbf{K}, \mathbf{V})$        ▷ Compute combined attention
22:   /* Split AttnOut for caching */
23:   $\mathbf{AttnOut}_p, \mathbf{AttnOut}_r \leftarrow \mathbf{AttnOut}_{1:|\mathbf{c}|}, \mathbf{AttnOut}_{|\mathbf{c}|+1:}$
24:   $\mathcal{C}_p[l][\texttt{attn}] \leftarrow \mathbf{AttnOut}_p$        ▷ Store prompt attention output
25:   $\mathcal{C}_r[l][\texttt{attn}] \leftarrow \mathbf{AttnOut}_r$        ▷ Store response attention output
26:   $\mathbf{h} \leftarrow \mathbf{x}_{in} + \mathbf{AttnOut}$        ▷ Post-attention residual
27:   /* --- FFN Block --- */
28:   $\mathbf{h}_{norm} \leftarrow \text{LayerNorm}(\mathbf{h})$
29:   $\mathbf{FFNOut} \leftarrow \text{FFN}(\mathbf{h}_{norm})$        ▷ Compute combined FFN output
30:   /* Split FFNOut for caching */
31:   $\mathbf{FFNOut}_p, \mathbf{FFNOut}_r \leftarrow \mathbf{FFNOut}_{1:|\mathbf{c}|}, \mathbf{FFNOut}_{|\mathbf{c}|+1:}$
32:   $\mathcal{C}_p[l][\texttt{mlp}] \leftarrow \mathbf{FFNOut}_p$        ▷ Store prompt FFN output
33:   $\mathcal{C}_r[l][\texttt{mlp}] \leftarrow \mathbf{FFNOut}_r$        ▷ Store response FFN output
34:   $\mathbf{x}_{out} \leftarrow \mathbf{h} + \mathbf{FFNOut}$        ▷ Final residual. Note: Code uses dropout here.
35:   $\mathbf{x}_{in} \leftarrow \mathbf{x}_{out}$        ▷ Update input for the next layer
36: **end for**
37: **return** $\mathcal{C}_p, \mathcal{C}_r$        ▷ Initialized caches

---

---

**Algorithm 3** dLLM-Cache: Case 1 - Full Refresh

---

**Require:** Layer input $\mathbf{x}_{in}$, layer index $l$, caches $\mathcal{C}_p$ and $\mathcal{C}_r$
$\qquad\qquad\qquad\qquad\qquad\qquad\qquad\qquad \triangleright \mathbf{x}_{in}$ is the output of layer $l-1$, or $[\mathbf{c}; \mathbf{y}^{(k)}]$ for $l=1$

1: /* Case 1:  Refresh both prompt and response */
2: /* --- Attention Block --- */
3: $\mathbf{x}_{norm} \leftarrow \text{LayerNorm}(\mathbf{x}_{in})$
4: $\mathbf{Q}, \mathbf{K}, \mathbf{V} \leftarrow \text{Q\_proj}(\mathbf{x}_{norm}), \text{K\_proj}(\mathbf{x}_{norm}), \text{V\_proj}(\mathbf{x}_{norm})$
5: /* Split K, V for caching */
6: $\mathbf{K}_p, \mathbf{K}_r \leftarrow \mathbf{K}_{1:|\mathbf{c}|}, \mathbf{K}_{|\mathbf{c}|+1:}$
7: $\mathbf{V}_p, \mathbf{V}_r \leftarrow \mathbf{V}_{1:|\mathbf{c}|}, \mathbf{V}_{|\mathbf{c}|+1:}$
8: $\mathcal{C}_p[l][\text{kv\_cache}] \leftarrow \{\mathbf{K}_p, \mathbf{V}_p\}$ $\qquad\qquad\qquad\qquad\quad \triangleright$ Update prompt KV cache
9: $\mathcal{C}_r[l][\text{kv\_cache}] \leftarrow \{\mathbf{K}_r, \mathbf{V}_r\}$ $\qquad\qquad\qquad\qquad\quad \triangleright$ Update response KV cache
10: $\mathbf{AttnOut} \leftarrow \text{Attention}(\mathbf{Q}, \mathbf{K}, \mathbf{V})$ $\qquad\qquad\qquad\quad \triangleright$ Compute combined attention
11: /* Split AttnOut for caching */
12: $\mathbf{AttnOut}_p, \mathbf{AttnOut}_r \leftarrow \mathbf{AttnOut}_{1:|\mathbf{c}|}, \mathbf{AttnOut}_{|\mathbf{c}|+1:}$
13: $\mathcal{C}_p[l][\text{attn}] \leftarrow \mathbf{AttnOut}_p$ $\qquad\qquad\qquad\qquad \triangleright$ Update prompt attention cache
14: $\mathcal{C}_r[l][\text{attn}] \leftarrow \mathbf{AttnOut}_r$ $\qquad\qquad\qquad\qquad \triangleright$ Update response attention cache
15: $\mathbf{h} \leftarrow \mathbf{x}_{in} + \mathbf{AttnOut}$ $\qquad\qquad\qquad\qquad\qquad\qquad \triangleright$ Post-attention residual
16: /* --- FFN Block --- */
17: $\mathbf{h}_{norm} \leftarrow \text{LayerNorm}(\mathbf{h})$
18: $\mathbf{FFNOut} \leftarrow \text{FFN}(\mathbf{h}_{norm})$ $\qquad\qquad\qquad\qquad \triangleright$ Compute combined FFN output
19: /* Split FFNOut for caching */
20: $\mathbf{FFNOut}_p, \mathbf{FFNOut}_r \leftarrow \mathbf{FFNOut}_{1:|\mathbf{c}|}, \mathbf{FFNOut}_{|\mathbf{c}|+1:}$
21: $\mathcal{C}_p[l][\text{mlp}] \leftarrow \mathbf{FFNOut}_p$ $\qquad\qquad\qquad\qquad\quad \triangleright$ Update prompt FFN cache
22: $\mathcal{C}_r[l][\text{mlp}] \leftarrow \mathbf{FFNOut}_r$ $\qquad\qquad\qquad\qquad\quad \triangleright$ Update response FFN cache
23: $\mathbf{x}_{out} \leftarrow \mathbf{h} + \mathbf{FFNOut}$ $\qquad\qquad\qquad\qquad\qquad\qquad \triangleright$ Final residual.
24: **return** $\mathbf{x}_{out}, \mathcal{C}_p, \mathcal{C}_r$ $\qquad\qquad\qquad \triangleright$ Return layer output and updated caches

---

**Algorithm 4** dLLM-Cache: Case 2 - Refresh Prompt Only

---

**Require:** Layer input $\mathbf{x}_{in}$, layer index $l$, caches $\mathcal{C}_p$ and $\mathcal{C}_r$
$\qquad\qquad\qquad\qquad\qquad\qquad\qquad\qquad\qquad\qquad \triangleright \mathbf{x}_{in}$ is the output of layer $l-1$

1: /* Case 2:  Refresh prompt only, reuse response features */
2: $\mathbf{x}_{p\_in} \leftarrow \mathbf{x}_{in,1:|\mathbf{c}|}$ $\qquad\qquad\qquad\qquad\qquad\qquad \triangleright$ Layer's prompt input part
3: /* Compute fresh prompt features */
4: $\mathbf{x}_{p\_norm} \leftarrow \text{LayerNorm}(\mathbf{x}_{p\_in})$
5: $\mathbf{Q}_p \leftarrow \text{Q\_proj}(\mathbf{x}_{p\_norm}); \mathbf{K}_p \leftarrow \text{K\_proj}(\mathbf{x}_{p\_norm}); \mathbf{V}_p \leftarrow \text{V\_proj}(\mathbf{x}_{p\_norm})$
6: $\mathcal{C}_p[l][\text{kv\_cache}] \leftarrow \{\mathbf{K}_p, \mathbf{V}_p\}$ $\qquad\qquad\qquad\qquad \triangleright$ Update prompt KV cache
7: /* Retrieve response features from cache */
8: $\{\mathbf{K}_r, \mathbf{V}_r\} \leftarrow \mathcal{C}_r[l][\text{kv\_cache}]$ $\qquad\qquad\qquad\qquad \triangleright$ Reuse cached response KV
9: /* Compute attention with mixed features */
10: $\mathbf{K} \leftarrow [\mathbf{K}_p; \mathbf{K}_r]; \mathbf{V} \leftarrow [\mathbf{V}_p; \mathbf{V}_r]$
11: $\mathbf{AttnOut}_p \leftarrow \text{Attention}(\mathbf{Q}_p, \mathbf{K}, \mathbf{V})$ $\qquad\qquad\qquad \triangleright$ Only compute prompt attention
12: $\mathcal{C}_p[l][\text{attn}] \leftarrow \mathbf{AttnOut}_p$ $\qquad\qquad\qquad\qquad \triangleright$ Update prompt attention cache
13: $\mathbf{AttnOut}_r \leftarrow \mathcal{C}_r[l][\text{attn}]$ $\qquad\qquad\qquad\qquad \triangleright$ Reuse cached response attention
14: $\mathbf{AttnOut} \leftarrow [\mathbf{AttnOut}_p; \mathbf{AttnOut}_r]$ $\qquad\quad \triangleright$ Combine prompt and response attention
15: $\mathbf{h} \leftarrow \mathbf{x}_{in} + \mathbf{AttnOut}$ $\qquad\qquad \triangleright$ Post-attention residual (using layer input $\mathbf{x}_{in}$)
16: /* --- FFN Block --- */
17: $\mathbf{h}_p, \mathbf{h}_r \leftarrow \mathbf{h}_{1:|\mathbf{c}|}, \mathbf{h}_{|\mathbf{c}|+1:}$ $\qquad\qquad\qquad\qquad\quad \triangleright$ Split post-attention state
18: $\mathbf{h}_{p\_norm} \leftarrow \text{LayerNorm}(\mathbf{h}_p)$
19: $\mathbf{FFNOut}_p \leftarrow \text{FFN}(\mathbf{h}_{p\_norm})$ $\qquad\qquad\qquad\qquad \triangleright$ Compute FFN for prompt
20: $\mathcal{C}_p[l][\text{mlp}] \leftarrow \mathbf{FFNOut}_p$ $\qquad\qquad\qquad\qquad \triangleright$ Update prompt FFN cache
21: $\mathbf{FFNOut}_r \leftarrow \mathcal{C}_r[l][\text{mlp}]$ $\qquad\qquad\qquad\qquad \triangleright$ Reuse cached response FFN
22: $\mathbf{FFNOut} \leftarrow [\mathbf{FFNOut}_p; \mathbf{FFNOut}_r]$ $\qquad\qquad\qquad \triangleright$ Combine FFN outputs
23: $\mathbf{x}_{out} \leftarrow \mathbf{h} + \mathbf{FFNOut}$ $\qquad\qquad\qquad\qquad \triangleright$ Final output for this layer
24: **return** $\mathbf{x}_{out}, \mathcal{C}_p, \mathcal{C}_r$ $\qquad\qquad\qquad \triangleright$ Return layer output and updated caches

---

---

**Algorithm 5** dLLM-Cache: Case 3 - Refresh Response Only

---

**Require:** Layer input $\mathbf{x}_{in}$, layer index $l$, caches $\mathcal{C}_p$ and $\mathcal{C}_r$

                                                                  $\triangleright$ $\mathbf{x}_{in}$ is the output of layer $l-1$

1: /* Case 3:  Refresh response only, reuse prompt features */

2: $\mathbf{x}_{r\_in} \leftarrow \mathbf{x}_{in,|\mathbf{c}|+1:}$           $\triangleright$ Layer's response input part

3: /* Retrieve prompt features from cache */

4: $\{\mathbf{K}_p, \mathbf{V}_p\} \leftarrow \mathcal{C}_p[l][\texttt{kv\_cache}]$        $\triangleright$ Reuse cached prompt KV

5: $\mathbf{AttnOut}_p \leftarrow \mathcal{C}_p[l][\texttt{attn}]$        $\triangleright$ Reuse cached prompt attention

6: $\mathbf{FFNOut}_p \leftarrow \mathcal{C}_p[l][\texttt{mlp}]$        $\triangleright$ Reuse cached prompt FFN

7: /* Compute fresh response features */

8: $\mathbf{x}_{r\_norm} \leftarrow \text{LayerNorm}(\mathbf{x}_{r\_in})$

9: $\mathbf{Q}_r \leftarrow \text{Q\_proj}(\mathbf{x}_{r\_norm}); \mathbf{K}_r \leftarrow \text{K\_proj}(\mathbf{x}_{r\_norm}); \mathbf{V}_r \leftarrow \text{V\_proj}(\mathbf{x}_{r\_norm})$

10: $\mathcal{C}_r[l][\texttt{kv\_cache}] \leftarrow \{\mathbf{K}_r, \mathbf{V}_r\}$        $\triangleright$ Update response KV cache

11: /* Compute attention with mixed features */

12: $\mathbf{K} \leftarrow [\mathbf{K}_p; \mathbf{K}_r]; \mathbf{V} \leftarrow [\mathbf{V}_p; \mathbf{V}_r]$

13: $\mathbf{AttnOut}_r \leftarrow \text{Attention}(\mathbf{Q}_r, \mathbf{K}, \mathbf{V})$        $\triangleright$ Only compute response attention

14: $\mathcal{C}_r[l][\texttt{attn}] \leftarrow \mathbf{AttnOut}_r$        $\triangleright$ Update response attention cache

15: $\mathbf{AttnOut} \leftarrow [\mathbf{AttnOut}_p; \mathbf{AttnOut}_r]$        $\triangleright$ Combine prompt and response attention

16: $\mathbf{h} \leftarrow \mathbf{x}_{in} + \mathbf{AttnOut}$        $\triangleright$ Post-attention residual (using layer input $\mathbf{x}_{in}$)

17: /* --- FFN Block --- */

18: $\mathbf{h}_p, \mathbf{h}_r \leftarrow \mathbf{h}_{1:|\mathbf{c}|}, \mathbf{h}_{|\mathbf{c}|+1:}$        $\triangleright$ Split post-attention state

19: /* Retrieve prompt FFN, Compute response FFN */

20: $\mathbf{h}_{r\_norm} \leftarrow \text{LayerNorm}(\mathbf{h}_r)$

21: $\mathbf{FFNOut}_r \leftarrow \text{FFN}(\mathbf{h}_{r\_norm})$        $\triangleright$ Compute FFN for response

22: $\mathcal{C}_r[l][\texttt{mlp}] \leftarrow \mathbf{FFNOut}_r$        $\triangleright$ Update response FFN cache

23: $\mathbf{FFNOut} \leftarrow [\mathbf{FFNOut}_p; \mathbf{FFNOut}_r]$        $\triangleright$ Combine FFN outputs

24: $\mathbf{x}_{out} \leftarrow \mathbf{h} + \mathbf{FFNOut}$        $\triangleright$ Final output for this layer

25: **return** $\mathbf{x}_{out}, \mathcal{C}_p, \mathcal{C}_r$        $\triangleright$ Return layer output and updated caches

---

---

**Algorithm 6** dLLM-Cache: Case 4 - Adaptive Update

---

**Require:** Layer input $\mathbf{x}_{in}$, layer index $l$, caches $\mathcal{C}_p$ and $\mathcal{C}_r$, adaptive update ratio $\rho$

1: /* Case 4:  Adaptive partial update when no refresh required */
2: /* Retrieve cached prompt features */
3: $\{\mathbf{K}_p, \mathbf{V}_p\} \leftarrow \mathcal{C}_p[l][\text{kv\_cache}]$
4: $\mathbf{AttnOut}_p \leftarrow \mathcal{C}_p[l][\text{attn}]$
5: $\mathbf{FFNOut}_p \leftarrow \mathcal{C}_p[l][\text{mlp}]$
6: **if** $\rho > 0$ **then**                ▷ Only proceed if adaptive update is enabled
7:     /* Compute current response Value projections */
8:     $\mathbf{x}_{r\_in} \leftarrow \mathbf{x}_{in,|\mathbf{c}|+1:}$              ▷ Layer's response input part
9:     $\mathbf{x}_{r\_norm} \leftarrow \text{LayerNorm}(\mathbf{x}_{r\_in})$
10:     $\mathbf{V}_r^{\text{new}} \leftarrow \text{V\_proj}(\mathbf{x}_{r\_norm})$
11:     /* Retrieve cached response features */
12:     $\{\mathbf{K}_r, \mathbf{V}_r\} \leftarrow \mathcal{C}_r[l][\text{kv\_cache}]$
13:     /* Compute similarity to identify tokens needing update */
14:     **for** each token index $j$ in response sequence **do**
15:         $s_j \leftarrow \frac{(\mathbf{V}_r^{\text{new}}[j])^\top \mathbf{V}_r[j]}{\|\mathbf{V}_r^{\text{new}}[j]\|\|\mathbf{V}_r[j]\|}$          ▷ Cosine similarity
16:     **end for**
17:     $\mathcal{I}_{\text{update}} \leftarrow$ indices of $\lfloor\rho|\mathbf{y}^{(k)}|\rfloor$ tokens with lowest $s_j$
18:     /* Selective computation for selected tokens */
19:     $\mathbf{x}_{r\_norm\_selected} \leftarrow$ gather tokens from $\mathbf{x}_{r\_norm}$ at indices $\mathcal{I}_{\text{update}}$
20:     $\mathbf{Q}_r^{\text{selected}} \leftarrow \text{Q\_proj}(\mathbf{x}_{r\_norm\_selected})$
21:     $\mathbf{K}_r^{\text{selected}} \leftarrow \text{K\_proj}(\mathbf{x}_{r\_norm\_selected})$
22:     /* Update KV cache with new values */
23:     $\mathbf{K}_r^{\text{updated}} \leftarrow \text{ScatterUpdate}(\mathbf{K}_r, \mathcal{I}_{\text{update}}, \mathbf{K}_r^{\text{selected}})$        ▷ Uses scatter
24:     $\mathcal{C}_r[l][\text{kv\_cache}] \leftarrow \{\mathbf{K}_r^{\text{updated}}, \mathbf{V}_r^{\text{new}}\}$        ▷ Always use new V
25:     /* Compute attention only for selected tokens */
26:     $\mathbf{K} \leftarrow [\mathbf{K}_p; \mathbf{K}_r^{\text{updated}}]; \mathbf{V} \leftarrow [\mathbf{V}_p; \mathbf{V}_r^{\text{new}}]$
27:     $\mathbf{AttnOut}_r^{\text{selected}} \leftarrow \text{Attention}(\mathbf{Q}_r^{\text{selected}}, \mathbf{K}, \mathbf{V})$
28:     /* Update response attention cache at selected positions */
29:     $\mathbf{AttnOut}_r \leftarrow \mathcal{C}_r[l][\text{attn}]$
30:     $\mathbf{AttnOut}_r^{\text{updated}} \leftarrow \text{ScatterUpdate}(\mathbf{AttnOut}_r, \mathcal{I}_{\text{update}}, \mathbf{AttnOut}_r^{\text{selected}})$
31:     $\mathcal{C}_r[l][\text{attn}] \leftarrow \mathbf{AttnOut}_r^{\text{updated}}$
32:     $\mathbf{AttnOut} \leftarrow [\mathbf{AttnOut}_p; \mathbf{AttnOut}_r^{\text{updated}}]$        ▷ Combine attn outputs
33:     $\mathbf{h} \leftarrow \mathbf{x}_{in} + \mathbf{AttnOut}$        ▷ Post-attention residual (using layer input $\mathbf{x}_{in}$)
34:     /* --- FFN Block (Adaptive) --- */
35:     $\mathbf{h}_p, \mathbf{h}_r \leftarrow \mathbf{h}_{1:|\mathbf{c}|}, \mathbf{h}_{|\mathbf{c}|+1:}$        ▷ Split post-attention state
36:     /* Gather tokens from response post-attention state */
37:     $\mathbf{h}_r^{\text{selected}} \leftarrow$ gather tokens from $\mathbf{h}_r$ at indices $\mathcal{I}_{\text{update}}$
38:     /* Compute FFN only for selected tokens */
39:     $\mathbf{h}_{r\_selected\_norm} \leftarrow \text{LayerNorm}(\mathbf{h}_r^{\text{selected}})$
40:     $\mathbf{FFNOut}_r^{\text{selected}} \leftarrow \text{FFN}(\mathbf{h}_{r\_selected\_norm})$
41:     /* Update response FFN cache at selected positions */
42:     $\mathbf{FFNOut}_r \leftarrow \mathcal{C}_r[l][\text{mlp}]$
43:     $\mathbf{FFNOut}_r^{\text{updated}} \leftarrow \text{ScatterUpdate}(\mathbf{FFNOut}_r, \mathcal{I}_{\text{update}}, \mathbf{FFNOut}_r^{\text{selected}})$
44:     $\mathcal{C}_r[l][\text{mlp}] \leftarrow \mathbf{FFNOut}_r^{\text{updated}}$
45:     $\mathbf{FFNOut} \leftarrow [\mathbf{FFNOut}_p; \mathbf{FFNOut}_r^{\text{updated}}]$        ▷ Combine FFN outputs
46: **else**                      ▷ Case: $\rho = 0$
47:     /* Pure cache retrieval - no updates */
48:     $\mathbf{AttnOut}_r \leftarrow \mathcal{C}_r[l][\text{attn}]$
49:     $\mathbf{AttnOut} \leftarrow [\mathbf{AttnOut}_p; \mathbf{AttnOut}_r]$
50:     $\mathbf{h} \leftarrow \mathbf{x}_{in} + \mathbf{AttnOut}$        ▷ Post-attention residual
51:     $\mathbf{FFNOut}_r \leftarrow \mathcal{C}_r[l][\text{mlp}]$
52:     $\mathbf{FFNOut} \leftarrow [\mathbf{FFNOut}_p; \mathbf{FFNOut}_r]$        ▷ Combine FFN outputs
53: **end if**
54: $\mathbf{x}_{out} \leftarrow \mathbf{h} + \mathbf{FFNOut}$        ▷ Final output for this layer
55: **return** $\mathbf{x}_{out}, \mathcal{C}_p, \mathcal{C}_r$        ▷ Return layer output and updated caches

---

