# OpenReview forum: "dLLM-Cache: Accelerating Diffusion Large Language Models with Adaptive Caching"
_ICLR.cc/2026/Conference — Submitted to ICLR 2026_

### Official Review · Reviewer_BFE1 · 2025-10-31

**Soundness:** 3
**Presentation:** 3
**Contribution:** 3
**Rating:** 8
**Confidence:** 5

**Summary:**

This paper introduces dLLM-Cache, a novel training-free framework designed to accelerate
inference in diffusion LLMs. It proposes a dual-component caching strategy. First, it
employs long-interval caching for the static input prompt, refreshing its features only
infrequently. Second, it uses an adaptive, short-interval caching mechanism for the
dynamically evolving response. This adaptive component is guided by a lightweight &quot;V-
verify&quot; mechanism. The paper demonstrates through extensive experiments on LLaDA and
Dream models that dLLM-Cache can achieve up to a 9.1x speedup, bringing dLLM inference
latency closer to that of autoregressive models without compromising output quality.

**Strengths:**

1. The paper addresses a critical and widely recognized problem: the high inference
cost of diffusion LLMs. The achieved speedups are substantial and represent a major
step towards making these models practical for real-world applications.

2. The V-verify mechanism is a clever, lightweight, and empirically-grounded solution
for adaptively selecting which tokens to update, avoiding the overhead of more
complex methods.
3. A key strength is that dLLM-Cache does not require any model retraining. This
makes the method easy to adopt.

**Weaknesses:**

1. The high-level diagrams provide a helpful overview of the caching workflow.
However, the specific structure of the cache, particularly whether it is a single global
entity or maintained on a per-layer basis, is not explicitly detailed.
2. The paper could better justify the necessity of caching four feature types (K, V,
AttnOut, FFNOut), as this is a departure from the more standard KV-caching in
ARMs.
3. It is unclear why some configurations, such as MMLU with LLaDA Instruct, benefit
less from caching than the Base model. An analysis for the variance in speedup
across different benchmarks may be beneficial.

**Questions:**

See weakness.

---

> ### Author Response · Authors · 2025-11-21
> **Author Response (Weakness 1, 2, 3):**
>
> Dear Reviewer BFE1,
>
> Many thanks to your valuable comments and questions, which help us a lot to improve our work. We address your questions as follows.
>
> ---
>
> >[W1] Clarify whether the cache is **global or layer-wise**.
>
> [A1] Thank you for the suggestion. The cache is managed on a **per-layer basis**. As detailed in Section 3.2 and illustrated in our Algorithm 2, for each Transformer layer $l$, we maintain separate caches for its prompt and response features.
>
> ---
>
> >[W2] Justify the **necessity of caching four feature types**.
>
> [A2] Thank you for this valuable comment. Our choice to cache four feature types: K, V, AttnOut, FFNOut is specific to dLLMs rather than ARMs. In dLLMs, all tokens are iteratively refined instead of generated sequentially, and most tokens show high feature stability across steps. Empirically, AttnOut and FFNOut **change minimally** for these stable tokens as shown in Figure 1. Therefore, **recomputing them is redundant**. We cache them selectively based on a similarity check (V-verify) and recompute only when significant changes occur. This design reduces redundant computation while maintaining model fidelity.
>
>
> ---
>
> >[W3] Explain the **variance** **in caching speedup** across different benchmarks and models.
>
> [A3] Thank you for the detailed observation, and we apologize for not clarifying this in the original manuscript. The variation in speedup ratios across models within the same task category arises from **differences in the total number of tokens and decoding steps** as shown in Appendix A.6. These differences result from our strict adherence to the settings used in the original LLaDA paper.
>
> For example:
> - **MMLU:** Base uses generation length 256 and 256 decoding steps, while Instruct uses only 3 for both.
> - **HumanEval:** Base uses 256 / 256; Instruct uses 512 / 512.
> - **Math:** Base uses 4-shot prompts, while Instruct uses 0-shot, leading to different prompt lengths.
>
> Under fixed configurations, the speedup remains stable and predictable. We have clarified this in the revised manuscript.

---

### Official Review · Reviewer_s8FB · 2025-11-02

**Soundness:** 3
**Presentation:** 3
**Contribution:** 2
**Rating:** 4
**Confidence:** 4

**Summary:**

This paper introduces a method for the caching mechanism in the diffusion large language models. The contribution of this paper includes the v-verify method to detect the tokens that show the most changed tokens for updating and then would not use the cache at those tokens. Experimental results show that the proposed method can achieve large inference acceleration compared to LLaDA and Dream.

**Strengths:**

1. The proposed V-Verify module effectively identifies and selects tokens for caching.
2. The method is well-motivated and demonstrates strong performance in accelerating dLLMs.

**Weaknesses:**

1. Contribution is limited

The paper claims two main contributions: (i) adopting different update intervals for the prompt and response, and (ii) introducing V-verify to identify the most changed tokens for partial updates. However, the first contribution, also emphasized in the main analysis experiments (Section 3.2), has already been explored in prior works on KV-Cache for diffusion LLMs, such as dKV-Cache (the prefill part) and Fast-dLLM (PrefixCache). The other contribution is the V-verify mechanism, which selects tokens to be cached based on the similarity between their V vectors. While this idea is interesting, it appears too incremental to constitute a sufficient contribution for a full paper.

2. Comparison with AR models requires stronger justification and additional results

The comparison with autoregressive models presented in Table 4 lacks clarity and fairness. The reported acceleration is marginal (47.73 vs 49.86 TPS), and the base model used for comparison is not instruction-tuned for reasoning tasks like GSM8K, making the results less meaningful. For instance, using a more appropriate baseline such as Llama3-8B-Instruct would yield around 79.6% accuracy on GSM8K, significantly higher than the 67.2% reported here. An apple-to-apple comparison is needed.

Furthermore, the procedure for measuring TPS and the experimental setup are unclear. From the FLOPs calculation in the appendix, it seems that diffusion LLMs inherently require more computation than AR models, so it is counterintuitive that they achieve higher speed. Clarification on this discrepancy and a more thorough comparison of speed with ARs is needed.

In addition, the efficiency evaluation is conducted with only 128 steps, resulting in relatively poor accuracy (67%) compared to the 71% and 78% reported in Table 1. A more comprehensive comparison across different steps is needed.

**Questions:**

1. In the v-verify stage, since q is not recalculated and only a subset of tokens is available for q, the input x to the attention input
x only has the hidden states with partial tokens. If we only have partial x/q/k/v vectors, how can the corresponding v vectors for the cached tokens be calculated as a new one?

2. Could you elaborate on why your approach outperforms dKV-Cache and Fast-dLLM? It would be helpful to provide some insights or for this improvement.

3. Is your method sensitive to the number of decoding steps? Given that many recent works support parallel decoding with multiple tokens per step and I think it would be a trend, how would your method perform when applied to those settings?

---

> ### Author Response · Authors · 2025-11-21
> **Author Response (Weakness 1, Question 2):**
>
> Dear Reviewer s8FB,
>
> Many thanks to your valuable comments and questions, which help us a lot to improve our work. We address your questions as follows.
>
> ---
>
> >[W1 Q2] Contribution is limited and elaboration on outperforming concurrent works
>
> [A1] Thank you for highlighting these related works. We would like to provide some context regarding the timeline of these publications. Our paper, along with dKV-Cache and Fast-dLLM, are the results of **concurrent and independent research**. **While we must strictly adhere to ICLR's double-blind policy** and cannot disclose the specific publication dates of each work, we feel it is important to clarify that **our work was, in fact, the earliest among these three to be completed and made publicly available**. We hope this context helps situate our contribution as a **pioneering exploration in this area, rather than an incremental improvement** on pre-existing methods.
>
> To explain why our method outperforms dKV-Cache and Fast-dLLM, it is useful to examine the limitations of their caching strategies. Fast-dLLM leverages the semi-autoregressive nature of dLLMs to avoid prefix re-computation via block-wise caching, which is a relatively coarse-grained strategy. However, it still requires the **full computational** path for the dynamic block currently being generated. dKV-Cache operates on the assumption that decoded token features stabilize immediately, employing a fixed-interval refresh mechanism. This **static approach fails to adapt to real-time dynamics**; even "decoded" tokens undergo internal feature shifts due to evolving bidirectional attention contexts, leading to accumulated errors and accuracy degradation.
>
> In contrast, our approach shifts from fixed heuristics to **feature-level adaptation**, deeply mining computational redundancy. We found that the **prompt is nearly static**, while the **response exhibits sparse, dynamic changes**, leading to our differentiated strategy of long-interval caching for prompts and short-interval, adaptive updates for responses. When handling the dynamic response, we move beyond fixed heuristics and introduce the **V-verify** mechanism as a **feature-level adaptation**. Its novelty lies in the key insight demonstrated in Figure 2: **the stability of the Value vector is the most reliable indicator of stability in downstream Attention and FFN outputs.** This enables a **data-driven, real-time** update policy that avoids the lag and inefficiency inherent in fixed schedules.
>
> Furthermore, unlike Fast-dLLM and dKV-Cache, which **cache only KV pairs**, our method extends caching to also cover **Attention and FFN outputs**. By applying independent adaptive updates at each layer, dLLM-Cache precisely identifies stable tokens and **bypasses their MLP computations**. Given that MLP layers account for the majority of parameters and computational cost, this fine-grained feature-level pruning delivers significantly higher speedups than simple KV caching without compromising generation quality.
>
> To illustrate these differences more clearly, we have summarized them in the table below:
>
> | Feature Comparison | dLLM-Cache (Ours) | dKV-Cache | Fast-dLLM  |
> | :-- | :--| :--| :--|
> | Core Mechanism| **Dual-component Adaptive Cache (Prompt/Response Differentiated)** | Delayed Caching| Block-wise Caching|
> | Cache Granularity  | **Feature-level (Fine-grained)**| Token-level (Coarse-grained) | Block-level (Coarse-grained) |
> | Update Trigger | **V-verify (Metric-driven, based on V-vector similarity)**  | Heuristic-based (on decoding state/delay) | Rule-based (on block boundaries) |
> | Key Insight| **Value vector is the best proxy for feature stability**| Decoded token features stabilize with a delay | In-block dependencies can be approximated |
> | Cached Components  | **KV Pairs, Attention Outputs, FFN Outputs**| KV Pairs Only | KV Pairs Only |

---

> > ### Author Response · Authors · 2025-11-21
> > **Author Response (Continued, W1, Q2)**
> >
> > Beyond the conceptual differences, the practical impact is most evident in the experimental results, as shown below.
> >
> > | **Task**  | **Method**              | **TPS↑**  | **Speed↑** | **Memory↓** | **Score↑** |
> > | :-------- | :---------------------- | :-------- | :--------- | :---------- | :--------- |
> > | MMLU      | LLaDA Instruct          | 10.12     | 1.00×      | 15.54       | 61.24      |
> > |           | + dKV-Cache             | 14.34     | 1.42×      | 17.88       | 60.87      |
> > |           | + Fast-dLLM             | 20.51     | 2.03×      | 17.13       | 61.43      |
> > |           | **+ dLLM-Cache (Ours)** | **21.23** | **2.10×**  | **16.61**   | **62.82**  |
> > | GSM8K     | LLaDA Instruct          | 6.95      | 1.00×      | 15.86       | 77.48      |
> > |           | + dKV-Cache             | 8.89      | 1.28×      | 21.08       | **79.30**  |
> > |           | + Fast-dLLM             | 19.11     | 2.75×      | 19.48       | 75.89      |
> > |           | **+ dLLM-Cache (Ours)** | **29.75** | **4.28×**  | **17.85**   | 78.54      |
> > | HumanEval | LLaDA Instruct          | 10.57     | 1.00×      | 15.39       | 38.71      |
> > |           | + dKV-Cache             | 14.40     | 1.36×      | 17.17       | 37.20      |
> > |           | + Fast-dLLM             | 21.50     | 2.03×      | **16.60**   | 36.59      |
> > |           | **+ dLLM-Cache (Ours)** | **44.77** | **4.24×**  | 16.65       | **39.02**  |
> >
> > As the results show, **our method achieves the most balanced and competitive performance** across benchmarks, consistently improving speed while preserving model quality. These findings indicate that the finer-grained design choices in our framework lead to practical advantages in real-world scenarios.
> >
> > We hope this additional clarification helps situate the contributions of our work. Taken together, the differentiated update strategy and the V-verify mechanism form a coherent and effective approach to efficient dLLM inference.

---

> ### Author Response · Authors · 2025-11-21
> **Author Response (Weakness 2):**
>
> >[W2.1] The comparison with AR models lacks fairness and needs an apple-to-apple comparison.
>
> [A2] Thank you for this constructive feedback.
>
> We would first like to clarify the primary goal of Table 4. As **an emerging research area**, the high inference latency of dLLMs is **a widely recognized bottleneck**, which is the core motivation for our work. Therefore, **the main purpose of Table 4 is to demonstrate the substantial improvement our method provides over the original dLLM baseline**. As shown, our dLLM-Cache boosts the throughput of LLaDA Base from 14.77 TPS to 49.15 TPS, achieving **a speedup of over 3.3×**. This successfully brings a model that was significantly slower than ARMs to a comparable speed level. Furthermore, our caching method has great potential as a fundamental acceleration technique, as it is **orthogonal to other dLLM acceleration methods like parallel sampling**. For instance, as shown in our paper, when combined with SlowFast Sampling, the final throughput (49.86 TPS) already surpasses that of the LLaMA3 Base model.
>
> We appreciate the reviewer's thoughtful point regarding model comparisons. In our initial experiments, we **followed the evaluation benchmark established in the original LLaDA paper**, which compared **LLaDA 8B Base against LLaMA3 8B Base**.
>
> That said, we fully agree that evaluating instruction-tuned models offers a more comprehensive perspective. Following this suggestion, we have conducted additional experiments comparing **LLaDA-8B-Instruct with LLaMA3-8B-Instruct on GSM8K**. The results are presented below:
>
> | Method                 | Steps | TPS↑      | Accuracy(%) on GSM8K |
> | :--------------------- | :---- | :-------- | :------------------- |
> | LLaMA3-8B-Instruct     | -     | 47.78     | 78.30                |
> | LLaDA-8B-Instruct      | 256   | 9.63      | 77.81                |
> | **+ Cache**            | 256   | 23.80 | **78.09**            |
> | **+ Cache + SlowFast** | -     | **49.94** | 77.26                |
>
> As the new results show, dLLM-Cache not only **raises the accuracy** of LLaDA Instruct to a level comparable to LLaMA3-8B-Instruct, but also **substantially accelerates inference**. When integrated with advanced parallel decoding strategies such as SlowFast Sampling, it delivers even **faster inference than ARMs**.
>
> These findings also highlight a broader point: dLLMs are still **an emerging paradigm**, and **only through more advances in acceleration**, such as **our caching mechanism**, can their speed advantage be fully realized and ultimately **surpass ARMs**.
>
> We hope this clarification and the additional experimental results have adequately addressed the points you raised.
>
> ---
>
> >[W2.2] Clarify the TPS measurement and explain the speed discrepancy between diffusion LLMs and AR models.
>
> [A3] Thank you for this insightful question. Our TPS metric is the **end-to-end generation speed**, calculated from the **wall-clock time** required to generate a complete sequence, reflecting true system performance.
>
> The phenomenon where dLLMs can achieve higher speeds despite higher theoretical FLOPs is due to **the different hardware bottlenecks that constrain each architecture's inference process**. During decoding, an AR model must read the entire set of model parameters from GPU memory to generate each single token. This results in a **very high memory access cost per generated token**. Because the amount of computation per step is small, the GPU's compute units are often idle while waiting for data, meaning the process is fundamentally limited by **memory bandwidth**, which is a classic "memory-bound" problem.
>
> In contrast, a diffusion LLM processes the entire sequence of length $L$ in parallel during each of its $K$ denoising steps. Although this requires $K$ forward passes, the total memory access cost is amortized across all $L$ tokens. The **average memory access cost per token for a dLLM is therefore significantly lower** than for an ARM (roughly $K/L$ times that of an ARM). Each step in a dLLM involves large-scale matrix computations that effectively saturate the GPU's compute resources. This makes the dLLM inference process "compute-bound", where performance is dictated by the GPU's floating-point operation capabilities.
>
> Therefore, even if a dLLM's total FLOPs are higher, its superior hardware utilization and lower amortized memory cost per token allow it to achieve a higher practical throughput (TPS) than an ARM that is severely constrained by memory bandwidth.

---

> > ### Author Response · Authors · 2025-11-21
> > **Author Response (Continued, W2)**
> >
> > >[W2.3] A more comprehensive comparison across different steps is needed.
> >
> > [A4] Thank you for this valuable suggestion. The higher accuracies of 71% and 78% reported in Table 1 correspond to the **LLaDA Base and Instruct** models evaluated at **256 denoising steps**, a setting we used to **ensure a strict and fair comparison with the original LLaDA paper**. The primary purpose of Table 4, on the other hand, was not to argue that dLLMs outperform ARMs in accuracy, but rather to showcase the **acceleration effect of our method over the dLLM baseline**, for which we chose 128 steps as a specific point of comparison.
> >
> > To make this comparison more comprehensive, we have added the results at 256 steps, as shown below:
> >
> > | **Method**         | **Steps** | **Throughput(TPS)↑** | **Accuracy(%)↑** |
> > | :----------------- | :-------- | :------------------- | :--------------- |
> > | LLaMA3 8B          | -         | 47.73                | 49.05            |
> > | LLaDA Base         | 256       | 7.32                 | 69.06            |
> > | + Cache            | 256       | 23.19                | 70.66            |
> > | LLaDA Base         | 128       | 14.77                | 64.14            |
> > | + Cache            | 128       | **49.15**            | 62.32            |
> > | + Cache + SlowFast | -         | **49.86**            | 67.17            |
> > | LLaDA Base         | 32        | 53.55                | 22.25            |
> >
> > Moreover, **Figure 4(c)** in our paper compares different steps and shows that directly reducing the steps indeed speeds up inference but **leads to a clear drop in performance**. In contrast, **dLLM-Cache** achieves substantial acceleration by removing redundant computations while **preserving high accuracy**.

---

> ### Author Response · Authors · 2025-11-21
> **Author Response (Question 1, 3):**
>
> > [Q1] How are new v vectors computed when only partial x/q/k/v tokens are available?
>
> [A5] Thank you for your thoughtful question. In the V-verify stage of dLLM-Cache, the mechanism does not operate with partial $V$ vectors. As described in Algorithm 6, the Adaptive Update step first **takes the full response input $x_{r,in}$** and computes **a complete set of new value vectors $V _ {r}^{new}$ through the $V\\_proj$ layer**. This full $V_{r}^{new}$ is then used in the V-verify mechanism, where it is compared against the previously cached $V_r$ vectors using cosine similarity. Based on this comparison, we identify a subset of tokens $\mathcal{I}_{update}$ that require updates.
>
> Only after this verification does the model compute new query and key vectors $Q _ {\text{selected}}$ and $K _ {\text{selected}}$ for the identified subset $\mathcal{I} _ {update}$. The $V$ cache is then fully replaced with the new $V _ {r}^{new}$, while the **$K$ cache is updated only at the indices in $\mathcal{I}_{update}$**. Finally, the attention mechanism uses the partial $Q_{selected}$ while accessing the full $K$ and $V$ caches.
>
> Therefore, the full $V_{r}^{new}$ is always computed during V-verify, which is essential to the mechanism. Although computing the full $V$ incurs some cost, it remains significantly lower than computing the full $Q$, $K$, attention output, and FFN output.
>
> ---
>
> >[Q3] Apply dLLM-Cache to parallel decoding methods.
>
> [A6] Thank you for this forward-looking question. Our method is designed to be **orthogonal** to such techniques, and is **not sensitive to the number of decoding steps**. While dLLM-Cache optimizes computation **within a single denoising step**, parallel decoding methods like SlowFast Sampling focus on **cross-step** sampling strategies. We believe these approaches are **complementary**. As detailed in Appendix A.2, our experiments show that combining dLLM-Cache with SlowFast Sampling **maintains strong performance across multiple benchmarks**. To further illustrate this synergy, we have **conducted additional experiments**, the results of which are presented below.
>
> | **Task**  | **Method**           | **TPS↑**  | **Speed(TPS)↑** | **Score↑** |
> | :-------- | :------------------- | :-------- | :-------------- | :--------- |
> | GSM8K     | LLaDA     | 4.55      | 1.00×           | 69.83      |
> |           | **+Cache +Sampling** | **26.99** | **5.93×**       | **69.60**  |
> | GPQA      | LLaDA     | 3.31      | 1.00×           | 31.47      |
> |           | **+Cache +Sampling** | **29.06** | **8.78×**       | **33.48**  |
> | Math      | LLaDA     | 5.14      | 1.00×           | 30.16      |
> |           | **+Cache +Sampling** | **26.50** | **5.16×**       | **29.42**  |
> | MMLU-pro  | LLaDA     | 9.16      | 1.00×           | 23.30      |
> |           | **+Cache +Sampling** | **33.38** | **3.64×**       | **25.53**  |
> | MMLU      | LLaDA     | 5.02      | 1.00×           | 62.11      |
> |           | **+Cache +Sampling** | **38.42** | **7.65×**       | **61.20**  |
> | BBH       | LLaDA     | 4.04      | 1.00×           | 44.97      |
> |           | **+Cache +Sampling** | **36.04** | **8.92×**       | **44.81**  |
> | MBPP      | LLaDA     | 4.98      | 1.00×           | 40.80      |
> |           | **+Cache +Sampling** | **27.26** | **5.47×**       | **39.00**  |
> | HumanEval | LLaDA     | 11.24     | 1.00×           | 31.71      |
> |           | **+Cache +Sampling** | **41.14** | **3.66×**       | **31.10**  |
>
> These results show that integrating dLLM-Cache **with parallel decoding methods** yields an **average 6x** speedup across various tasks while largely **maintaining model performance**, confirming its generality and potential as a core component for future dLLM acceleration.

---

> ### Author Response · Authors · 2025-11-28
>
> Dear Reviewer s8FB,
>
> Thank you once again for your valuable comments on our submission. As the discussion phase is approaching its end, we would like to kindly confirm whether we have sufficiently addressed all of your concerns (or at least part of them). Should there be any remaining questions or areas requiring further clarification, please do not hesitate to let us know. If you are satisfied with our responses, we would greatly appreciate your consideration in adjusting the evaluation scores accordingly. We sincerely look forward to your feedback.
>
> Best regards, Author Team

---

### Official Review · Reviewer_S22Y · 2025-11-04

**Soundness:** 3
**Presentation:** 3
**Contribution:** 3
**Rating:** 4
**Confidence:** 3

**Summary:**

This paper introduces dLLM-Cache, an adaptive caching framework designed to accelerate diffusion-based large language models (dLLMs) without retraining. The key idea is to cache intermediate representations during inference, leveraging redundancy in prompts and generated tokens. The proposed cache selection strategy dynamically balances accuracy and efficiency using prompt similarity and token reuse mechanisms. Experiments on various LLM-based diffusion tasks (e.g., LLaDA, DreamFusion) show that dLLM-Cache achieves substantial inference speedup with minimal degradation in performance.

**Strengths:**

1. The paper is clearly written and well-structured, making the method easy to understand.
2. The adaptive caching mechanism is simple yet effective, requiring no retraining or architectural modification.
3. Comprehensive experiments demonstrate strong empirical improvements across multiple diffusion-based LLMs.
4. Ablation studies support the robustness of the cache selection mechanism.

**Weaknesses:**

1. The theoretical complexity reduction is not analyzed in sufficient depth. A more formal discussion or derivation of the speed–accuracy trade-off would greatly enhance clarity.
2. Since dLLM-Cache reuses outdated KV pairs, it would strengthen the work to mathematically and empirically analyze the upper bound of the approximation error under different values of K.
3. The limitations of the proposed method, particularly under dynamic or semantically diverse prompts, are not fully explored. A deeper investigation of such edge cases would improve the completeness of the study.
4. Several tables (e.g., Table 1, Table 2, and Table 3) present overlapping or repetitive results, with only minor variations in metrics or experimental settings. These could be merged or reorganized to improve readability and reduce redundancy.
5. The observation that using outdated cache can sometimes improve accuracy is intriguing but insufficiently discussed. A more detailed explanation or hypothesis about why this occurs would make the findings more convincing.

**Questions:**

N/A

---

> ### Author Response · Authors · 2025-11-21
> **Author Response (Weakness 1):**
>
> Dear Reviewer S22Y,
>
> Many thanks to your valuable comments and questions, which help us a lot to improve our work. We address your questions as follows.
>
> ---
> > [W1] Need a more formal discussion or derivation of the speed–accuracy trade-off.
>
> [A1] Thank you for this insightful comment. A complete, closed-form theoretical derivation of the speed-accuracy trade-off is highly challenging, as accuracy is an empirical outcome that depends on the model, the task, and the approximation error introduced by our caching strategy, rather than being an analytical variable in a complexity formula. To ground the discussion in what can be theoretically characterized, our analysis begins with the computational complexity of a standard dLLM, formalized as $\text{FLOPs}_{\text{dLLM}} = K \cdot T \cdot (8nd^2 + 4n^2d + 4ndm)$. The critical bottleneck here is the term $4n^2d$, which arises from the **bidirectional self-attention mechanism** operating over the entire sequence of length $n$. This quadratic dependency on sequence length, **repeated across $K$ denoising steps**, is the primary reason for the high inference latency of dLLMs. Our framework is precisely designed to mitigate the cost of this dominant term.
>
> In contrast, the complexity of our method **dLLM-Cache** can be approximated as:
>
> $$
> \begin{aligned}
> \text{FLOPs}_{\text{dLLM-Cache}} \approx\
> &\frac{K}{K_p} \cdot T \cdot (8nd^2 + 4n^2d + 4ndm) \\\\
> &+ \left(\frac{K}{K_r} - \frac{K}{K_p}\right) \cdot T \cdot (8rd^2 + 4rnd + 4rdm) \\\\
> &+ K \cdot \left(1 - \frac{1}{K_r}\right) \cdot T \cdot (8\hat{r}d^2 + 4\hat{r}nd + 4\hat{r}dm)
> \end{aligned}
> $$
>
> It is a weighted sum of three distinct operational modes. The **first term**, $\frac{K}{K_p} \cdot T \cdot (\dots)$, represents the cost of periodic full refreshes. While its per-instance cost is identical to the original dLLM, its frequency is low, controlled by a large $K_p$(often ≥ 100).
>
> The **second term**, $(\frac{K}{K_r} - \frac{K}{K_p}) \cdot T \cdot (8rd^2 + 4rnd + 4rdm)$, quantifies the cost of the more frequent **response-only refreshes**. Here, the significant gain becomes apparent: the quadratic attention term is reduced from $O(n^2)$ to $O(rn)$, as we only compute new query vectors for the response tokens (length $r$) to attend to the full sequence (length $n$). This term represents a middle ground, ensuring the evolving response is updated regularly while leveraging a static cache for the prompt.
>
> The **third and final term**, $K \cdot (1 - \frac{1}{K_r}) \cdot T \cdot (8\hat{r}d^2 + 4\hat{r}nd + 4\hat{r}dm)$, is **the cornerstone of our efficiency**. This term accounts for the **adaptive partial updates**, which constitute the vast majority of the denoising steps. The computational load is further minimized by replacing $r$ with $\hat{r} = \rho \cdot r$, where $\rho$ is typically a small fraction like 25%. This reflects our core insight that only a sparse subset of response tokens requires updating at each step. This operation, while being the most frequent, is by far the computationally cheapest.
>
> This analysis clarifies the inherent **speed-accuracy trade-off** governed by our key hyperparameters. Increasing the cache intervals $K_p$, $K_r$ or lowering the update ratio $\rho$ directly boosts theoretical speedup by reducing costly computations, but risks using stale states that may degrade output quality. **Due to the non-linear nature of generation, a precise analytical formula for this trade-off is intractable.** Therefore, we adopted a rigorous empirical approach. Our ablation studies in **Figure 4 and Figure 5** systematically map this relationship, identifying configurations that achieve significant acceleration while preserving model fidelity.
>
> We hope this more detailed explanation clarifies the theoretical underpinnings of our method's efficiency and its inherent trade-offs.

---

> ### Author Response · Authors · 2025-11-21
> **Author Response (Weakness 2):**
>
> > [W2] Analyze the upper bound of the error under different values of K.
>
> [A2] Thank you for raising this critical point. While **deriving a strict, closed-form error bound for a complex, iterative system like a Transformer-based model is exceptionally challenging**, our goal here is to formally demonstrate that the error introduced by dLLM-Cache is **rigorously bounded** and to elucidate the mechanisms that ensure this stability.
>
> Our analysis begins by formalizing the error source. Let $y_k$ be the true hidden state trajectory and $\tilde{y} _ k$ be the approximated trajectory. The error at step $k$ is $$\delta_k = y _ k - \tilde{y} _ k$$. This error propagates through the Transformer network, $F$, which we assume to be Lipschitz continuous with a constant $L_F$. The denoising update can be abstracted as $$y _ {k-1} \approx \alpha_k y_k + (1-\alpha_k)F(y_k)$$. The error at the next step, $\delta_{k-1}$, can then be bounded as follows:
>
> $$
> \\begin{aligned}
> \\|\\delta_{k-1}\\| &= \\|(\\alpha_k y_k + (1-\\alpha_k)F(y_k)) - (\\alpha_k \\tilde{y}_k + (1-\\alpha_k)F(\\tilde{y}_k))\\| \\\\
> &\\le \\alpha_k \\|\\delta_k\\| + (1-\\alpha_k)\\|F(y_k) - F(\\tilde{y}_k)\\| \\\\
> &\\le \\alpha_k \\|\\delta_k\\| + (1-\\alpha_k)L_F \\|\\delta_k\\| = (\\alpha_k + (1-\\alpha_k)L_F) \\|\\delta_k\\|
> \\end{aligned}
> $$
>
> Letting $$C_k = \alpha_k + (1-\alpha_k)L_F$$, we have $$\|\delta_{k-1}\| \le C_k \|\delta_k\|$$. Since $L_F > 1$ for expressive models, $C_k > 1$, indicating a natural tendency for the error to amplify exponentially without intervention.
>
> This is where the dual-mechanism design of dLLM-Cache becomes critical. The first mechanism is the **periodic error reset** enforced by the response refresh interval, $K_r$. At every refresh step $k_0$ (where $k_0 \pmod{K_r} = 0$), the error is effectively reset, i.e., $\|\delta_{k_0}\| \approx 0$. This guarantees that the error accumulation is confined within finite windows of length $K_r$. If error were to accumulate unchecked for $j$ steps from $k_0$, its magnitude would be roughly $$\|\delta_{k_0-j}\| \le (\prod_{i=1}^j C_{k_0-i+1})\|\delta_{k_0}\|$$. The reset mechanism prevents $j$ from growing indefinitely, thus ensuring the overall error is bounded.
>
> However, to further control the error within these intervals, dLLM-Cache employs the **V-verify adaptive update**. At each non-refresh step, V-verify updates a fraction $\rho$ of the tokens. This means only the error corresponding to the cached $(1-\rho)$ fraction of tokens, let's denote its projection by $P_{cache}$, is amplified. The error recursion is more accurately described as:
> $$
> \|\delta_{k-1}\| \le C_k \|P_{cache}(\delta_k)\| + \epsilon_{\text{step}}
> $$
> where $\epsilon_{\text{step}}$ is the small error from the newly computed tokens. Since $\|P_{cache}(\delta_k)\|$ is strictly smaller than $\|\delta_k\|$, this leads to a smaller **effective error amplification factor**, $C'_k < C_k$. Consequently, the maximum error accumulated within a refresh window is significantly reduced. Instead of peaking at an order of $O(C^{K_r})$, **the error peak is now on the order of $O((C')^{K_r})$, which is a much tighter bound**.
>
> In conclusion, our mathematical analysis demonstrates that the approximation error in dLLM-Cache is robustly bounded. **The upper bound $\sup_k |\delta_k|$ depends on the model’s properties through $C_k$, on the refresh interval $K_r$, and on the adaptive update ratio $\rho$, which determines $C'_k$.** The periodic reset $K_r$ provides a hard guarantee of boundedness by preventing infinite error accumulation, while the V-verify significantly tighten this bound by actively suppressing error growth at almost every step.
>
> **This theoretical finding aligns perfectly with our experimental observations**, as shown in **Figure 4(b)**. The configuration without V-verify ($\rho$=0, gray triangular line) exhibits a significant decline in accuracy as $K_r$ increases. In contrast, the configuration with **V-verify** enabled (even when updating only 25% of tokens, $\rho$=0.25, orange triangular line) **maintains nearly the same accuracy as the baseline**, even at larger $K_r$ values that yield substantial computational savings. This demonstrates that the considerable acceleration achieved by our method, without sacrificing high fidelity, is **attributable to well-controlled errors**.

---

> ### Author Response · Authors · 2025-11-21
> **Author Response (Weakness 3, 4):**
>
> >[W3] Explore the limitations under dynamic or semantically diverse prompts.
>
> [A3] Thank you for raising this point. To better investigate the limitations of our method under dynamic or semantically diverse prompts, we have conducted extensive additional experiments using the **LongBench**[1] benchmark, evaluating both **LLaDA Instruct** and **Dream Instruct**.
>
> LongBench covers **six major task categories**: single-doc QA, multi-doc QA, summarization, few-shot learning, synthetic tasks, and code completion. The texts in LongBench are **notably long**, averaging about 6,711 words for English tasks and approximately 13,386 characters for Chinese tasks. These tasks **exhibit high semantic complexity and structural diversity**, making them well-suited to **represent real-world scenarios with semantic diversity**. By evaluating our dLLM-Cache on LongBench, we are able to more comprehensively examine its performance on such edge cases. The detailed results are as follows:
>
>
> **Table 1: Detailed LongBench Results for LLaDA Instruct.**
>
> | Category | Task | w/o Cache | w/ Cache |
> | :--- | :--- | :---: | :---: |
> | **Single-Doc. QA** | Qasper | 16.96 | 15.26 |
> | | MF-en | 31.31 | 29.62 |
> | **Multi-Doc. QA** | HotpotQA | 14.68 | 13.87 |
> |  | 2WikiMQA | 17.60 | 17.17 |
> |  | Musique | 11.48 | 10.44 |
> | **Summarization** | GovReport | 29.24 | 29.75 |
> |  | QMSum | 21.93 | 22.06 |
> |  | MultiNews | 27.58 | 26.68 |
> | **Few-shot Learning** | TREC | 65.20 | 66.00 |
> |  | TriviaQA | 47.98 | 44.94 |
> |  | SAMSum | 40.51 | 41.86 |
> | **Synthetic** | PRe | 98.17 | 97.44 |
> |  | Lcc | 65.69 | 66.07 |
> | **Code** | RB-P | 59.57 | 59.34 |
> | | **Ave. Score** | **39.14** | **38.61** |
>
> **Table 2: Detailed LongBench Results for Dream Instruct.**
>
> | Category              | Task           | w/o Cache | w/ Cache  |
> | :-------------------- | :------------- | :-------: | :-------: |
> | **Single-Doc. QA**    | Qasper         |   28.17   |   26.55   |
> |                       | MF-en          |   36.23   |   39.86   |
> | **Multi-Doc. QA**     | HotpotQA       |   27.65   |   27.66   |
> |                       | 2WikiMQA       |   32.43   |   32.09   |
> |                       | Musique        |   11.83   |   11.12   |
> | **Summarization**     | GovReport      |   5.04    |   4.40    |
> |                       | QMSum          |   14.29   |   13.89   |
> |                       | MultiNews      |   5.95    |   5.51    |
> | **Few-shot Learning** | TREC           |   73.00   |   73.50   |
> |                       | TriviaQA       |   89.25   |   89.59   |
> |                       | SAMSum         |   37.84   |   36.07   |
> | **Synthetic**         | PRe            |   16.92   |   12.05   |
> |                       | Lcc            |   38.91   |   39.88   |
> | **Code**              | RB-P           |   45.08   |   45.57   |
> |                       | **Ave. Score** | **33.04** | **32.70** |
>
> As demonstrated by the results above, our caching strategy **maintains robust performance** across various challenging tasks without impeding the model's reasoning capability over long and semantically complex prompts. We believe **these comprehensive experiments effectively address the concerns regarding methodological limitations** and substantially strengthen the completeness of our study.
>
>
> ---
>
> >[W4] Tables could be merged or reorganized.
>
> [A4] Thank you for the suggestion regarding table organization. The current separation of Tables 1 and 2 from Table 3 reflects two distinct analytical perspectives. Tables 1 and 2 focus on the internal validation of dLLM-Cache, where detailed efficiency metrics such as TPS and FLOPs are central to evaluating the improvements over the original baselines. Table 3, in contrast, provides a comparison with concurrent external methods, where emphasizing TPS, memory usage, and overall score offers a clearer view of practical performance differences. Merging these analyses into a single table would introduce heterogeneous metrics and settings, which may obscure the intent of each experiment and reduce readability. We truly appreciate your concern, and we will add brief clarifications in our manuscript before each table to better communicate their respective purposes and improve the overall flow of the experimental section.
>
> ---
> [1] LongBench: A Bilingual, Multitask Benchmark for Long Context Understanding (Bai et al., ACL 2024)

---

> ### Author Response · Authors · 2025-11-21
> **Author Response (Weakness 5):**
>
> >[W5] Explain or hypothesize the reason behind the observed accuracy improvement.
>
> [A5] Thank you for highlighting this observation. After careful consideration and analysis of the iterative denoising process in dLLMs, we propose a hypothesis centered around the concepts of **implicit regularization** and **denoising trajectory stabilization**.
>
> The generation process of a dLLM is **not a direct, one-shot mapping** but **an iterative refinement process**, akin to solving an optimization problem. Starting from a fully masked sequence with high entropy, the model progressively reduces uncertainty at each step to converge towards a high-likelihood output. In such a dynamic system, each intermediate step's prediction contains a degree of noise and uncertainty. A model that **reacts too sensitively to this transient noise at every single step might "overfit" to ephemeral**, non-optimal states, leading its generation path, what we term the "denoising trajectory", to oscillate or get trapped in suboptimal local minima of the vast semantic space.
>
> Our hypothesis is that using a moderately outdated cache acts as a form of **implicit temporal regularization**. By preserving the representations of most tokens for several steps, with static prompts maintained through a large $K_p$ and stable response tokens ensured by V-verify, dLLM-Cache introduces an element of "inertia" or "momentum" into the generation process. This mechanism prevents the model from making drastic changes based on the slight, noisy variations in each step's prediction. Instead, it is encouraged to find a more stable and generalized representation of the context, effectively smoothing out the high-frequency oscillations in the denoising trajectory. This is analogous to momentum in optimization algorithms, which helps to navigate past small bumps and noisy gradients to find a better, more global optimum.
>
> Furthermore, this phenomenon can be viewed through the lens of the **exploration-exploitation trade-off**. A full, precise recalculation at every step represents a pure "exploitation" strategy, where the model fully trusts its current, potentially noisy, prediction. Using an outdated cache introduces a subtle, controlled form of "exploration." It forces the model to make predictions based on a slightly perturbed or "lagged" view of the context. This slight discrepancy might prevent the model from prematurely committing to a specific semantic path that seems locally optimal but is globally suboptimal. By being forced to be robust to these minor temporal inconsistencies in its own internal state, the model may be guided towards a more robust and, in some cases, more accurate final output.
>
> In summary, we suggest that the observed accuracy improvements may partly arise from our caching mechanism acting as an **implicit regularizer**. By stabilizing the denoising trajectory and moderating the exploration-exploitation trade-off, the method might help prevent overfitting to intermediate noise, occasionally guiding the model toward a better solution. This is a tentative hypothesis, as the minor accuracy gains could also result from the inherent stochasticity of the generative process. Nonetheless, we find this interpretation intriguing and believe it points to a promising direction for future research on the regularization effects of caching strategies.

---

> ### Comment · Reviewer_S22Y · 2025-11-26
>
> Thank you for your answers. Most of my concerns have been addressed. After carefully reviewing the numbers in the table, I noticed that some results reported in this paper are lower than those in the LLaDA paper. For example, for LLaDA-Instruct on MMLU, this paper reports an accuracy of 61.24, whereas the LLaDA paper reports 65.5. On HumanEval, the numbers are 38.71 vs. 47.6. Could you please explain these discrepancies?

---

> ### Author Response · Authors · 2025-11-28
> **Response to Discrepancies Between Our Results and the Original LLaDA Paper**
>
> Thank you for your feedback and for your meticulous review of our experimental results.
>
> Regarding the score differences between our reproduced results and those reported in the original LLaDA paper, we would like to provide the following clarification:
>
> **At the time we conducted our experiments and prepared this submission, the LLaDA authors had not yet open-sourced their official evaluation framework for LLaDA-8B-Instruct.** Consequently, we implemented the evaluation pipelines for benchmarks such as MMLU and HumanEval **independently** to establish the baselines.
>
> As is well known in the LLM evaluation community, even minor differences in implementation details such as prompt templates, few-shot exemplars, temperature settings, or output parsing logic can lead to noticeable variations in absolute scores. This is the primary reason for the numerical differences between our reproduced baseline and the numbers reported in the original LLaDA paper.
>
> However, we emphasize that **we applied the exact same evaluation setup** (including identical prompts, sampling parameters, and post-processing logic) to **both the LLaDA baseline and our dLLM-Cache method**. Therefore, the relative performance comparison remains completely fair, and the effectiveness of our proposed method is fully valid.
>
> To further address your concern with the most up-to-date evidence, after the LLaDA authors recently open-sourced their official evaluation codebase, we re-evaluated both the original LLaDA-8B-Instruct model and our dLLM-Cache under their **official evaluation framework**. The results are as follows:
>
> | Task       | Method     | Speed↑ | Score↑        |
> |------------|------------|--------|----------------|
> | MMLU       | LLaDA-8B-Instruct  | 1.00×  | 65.3       |
> |            | + dLLM-Cache    | **2.00×**  | **65.3**|
> | HumanEval  | LLaDA-8B-Instruct  | 1.00×  | 46.8       |
> |            | + dLLM-Cache    | **4.00×**  | **46.6**|
>
>
> As shown in the table, under the official LLaDA evaluation protocol:
> - dLLM-Cache **preserves the performance** of the original LLaDA model under the official evaluation protocol.
> - dLLM-Cache provides a **4.0×** end-to-end speedup on HumanEval and a **2.0×** speedup on MMLU.
>
> We hope this explanation, together with the new results from the official evaluation framework, fully resolves your concern.
>
> Thank you again for your careful review and valuable feedback.

---

### Author Response · Authors · 2025-11-21
**General Response to Area Chair and Reviewers**

Dear Area Chair and Reviewers,

We explicitly thank the Area Chair and all reviewers (*Reviewers S22Y, s8FB, BFE1*) for their time and constructive feedback. We are encouraged by the reviewers’ shared recognition of the importance of improving the efficiency of diffusion LLMs and by their positive assessment of our proposed approach.

We especially appreciate the acknowledgment that high inference cost remains a "**critical and widely recognized problem**" (*Reviewer BFE1*). We would like to gently reiterate that diffusion LLMs are still **in an early stage** compared with mature autoregressive models (ARMs). Progress in acceleration, such as **the caching strategy explored in this work**, is **essential for realizing their potential parallelism**. Continued advances along this direction will be key to enabling dLLMs to fully demonstrate their latency benefits and eventually surpass ARMs.

Regarding the methodology, we value the consensus on the **training-free** and **efficient nature** of our design. Reviewers noted that dLLM-Cache is "easy to adopt" (*Reviewer BFE1*) without requiring architectural modifications (*Reviewer S22Y*). Specifically, the **V-Verify mechanism** was highlighted as "clever, lightweight, and empirically grounded" (*Reviewer BFE1*) and "effective" in identifying tokens for caching (*Reviewer s8FB*). This design enables substantial performance gains, with "comprehensive experiments" showing up to **9.1x inference speedup** without compromising output quality.

We have carefully addressed specific concerns regarding theoretical analysis, baseline comparisons, and technical details in our individual responses below. We have also updated our manuscript to reflect these improvements. We hope these clarifications convey our commitment to rigor and help position dLLM-Cache as an **early but meaningful** step toward making dLLMs a practical alternative to ARMs.

We are sincerely grateful once again for your feedback. If there are any further questions or comments, we would be glad to continue the discussion and will do our best to address them.

Thank you, Authors

---

### Meta-Review · Area_Chair_gqBx · 2026-01-06

**Summary:**

This paper proposes dLLM-Cache, a training-free caching framework that accelerates diffusion LLM inference by up to 9.1× through adaptive prompt and response caching guided by a V-verify mechanism. One reviewer flagged significant novelty limitations, noting substantial overlap with concurrent work (dKV-Cache, Fast-dLLM) and questioning whether the V-verify mechanism alone provides sufficient contribution for a full paper. Additionally, Reviewer S22Y's assessment contains factual errors and was flagged by an AI-detection service and the reviewer also acknowledged insufficient familiarity with the topic; we therefore discount this review from consideration. With only two remaining valid reviews expressing divergent opinions, the paper suffers from insufficient review coverage while key concerns about incremental contribution remain unresolved. Future work would benefit from clearer positioning relative to concurrent methods and deeper technical analysis of the V-verify mechanism's contributions. Based on these considerations, I recommend rejecting this submission.

**Reviewer Concerns:**

Addressed: Authors provided detailed comparison with concurrent works, additional experimental validation, and technical analysis of the V-verify mechanism.

Outstanding: Incremental novelty relative to concurrent work (dKV-Cache, Fast-dLLM) remains a core concern. One reviewer questions whether V-verify mechanism alone constitutes sufficient contribution. More critically, with only two valid reviews (one strongly positive at 8, one critical at 4), the paper lacks sufficient review coverage for confident decision-making.

**Reviewer Scores:**

BFE1 (8): Would maintain 8; positive on practical efficiency gains
S22Y (4): Discounted due to AI-detection flagging, factual errors, and self-acknowledged insufficient familiarity; would not factor into decision
s8FB (4): Would maintain 4; novelty concerns persist despite rebuttal

---

### Decision · Program_Chairs · 2026-01-26

Reject